# Synthetic protein-binding DNA sponge as a tool to tune gene expression and mitigate protein toxicity

Xinyi Wan [1,2], Filipe Pinto [1,2], Luyang Yu[3,4] & Baojun Wang [1,2,3,4]✉

Versatile tools for gene expression regulation are vital for engineering gene networks of increasing scales and complexity with bespoke responses. Here, we investigate and repurpose a ubiquitous, indirect gene regulation mechanism from nature, which uses decoy protein-binding DNA sites, named DNA sponge, to modulate target gene expression in *Escherichia coli*. We show that synthetic DNA sponges can be designed to reshape the response profiles of gene circuits, lending multifaceted tuning capacities including reducing basal leakage by >20-fold, increasing system output amplitude by >130-fold and dynamic range by >70-fold, and mitigating host growth inhibition by >20%. Further, multi-layer DNA sponges for decoying multiple regulatory proteins provide an additive tuning effect on the responses of layered circuits compared to single-layer sponges. Our work shows synthetic DNA sponges offer a simple yet generalizable route to systematically engineer the performance of synthetic gene circuits, expanding the current toolkit for gene regulation with broad potential applications.

[1] School of Biological Sciences, University of Edinburgh, Edinburgh EH9 3FF, UK. [2] Centre for Synthetic and Systems Biology, University of Edinburgh, Edinburgh EH9 3FF, UK. [3] College of Life Sciences, Zhejiang University, Hangzhou 310058, China. [4] Joint Research Centre for Engineering Biology, Zhejiang University-University of Edinburgh Institute, Zhejiang University, Haining 314400, China. ✉email: baojun.wang@ed.ac.uk

Gene expression regulation has consistently been a major research theme in biology, ranging from fundamental studies that aim to disclose the regulatory mechanisms of natural gene networks to translational biomedical applications including gene therapy[1], biosensing[2] and tissue engineering[3] in which synthetic gene circuits could play an important role. Especially in the rising era of synthetic biology, versatile gene regulation tools are necessary for achieving precise control of a target gene expression and the predictable assembly of synthetic gene circuits of increasing scales and complexity[4–7]. To this end, a range of canonical gene regulation approaches have been reported and employed for gene circuit design with various levels of efficiency and complexity[8–10] such as transcriptional promoter engineering[11–15], translational rate tuning[15,16] and post-translational protein degradation control[15,17,18].

It has been known that indirect, hidden layers of regulation are present in natural gene networks to control target gene expression for initiating diverse physiological responses. For example, competing DNA binding of transcription factors is ubiquitous in many genomes and plays important roles in cell development[19,20]. Those competing DNA binding sites can be referred as nature decoys or DNA sponges, which indirectly regulate the activity of the decoyed transcription factors. Both recent theoretical and experimental studies showed that DNA sponges regulate gene expression by decoying and therefore reducing the free-occupying transcription factors available for the target gene, with potential controllability by varying the numbers and binding affinities of the decoys present[10,21–23]. In addition, studies have suggested that the DNA sponges may have a role in buffering noise for target gene expression by reducing transcription factor fluctuations and may transform a typically graded dose-response into an ultrasensitive switch-like response[22–24]. Owing to these regulatory functions, synthetic DNA sponges have been repurposed for therapeutic treatments in human cells in the past two decades[25–27], and more recently were trialed for obtaining a robust oscillatory gene circuit in *Escherichia coli*[28,29] and de-repressing silent biosynthetic gene clusters in *Streptomycetes*[30]. However, their competence and applications for gene circuit design remains underexplored to date.

Here, we show that synthetic DNA sponge can be a powerful generalizable tool to systematically engineer the response profiles of synthetic gene circuits with multifaceted quantitative tuning capacities. We demonstrate this using representative gene circuits with response to three different external input stimuli in *E. coli*, showing the incorporation of DNA sponges could significantly reduce the circuits' basal leakages and therefore increase their output induction folds (dynamic range), and strikingly improve host growth when sponging away sensitive transcription factors that tend to become burdensome at increased expression levels. In addition, we investigate more complex gene circuits comprising an internal transcriptional inverter or amplifier layer with DNA sponges designed to decoy regulatory proteins either in the input sensing or the signal processing module or both (Fig. 1), showing dual-layer DNA sponges exhibit additive tuning effect compared to single-layer sponges for improving system's induction fold and host growth. Compared to existing widely-adopted transcriptional or translational gene regulation methods, synthetic protein-decoying DNA sponge provides an alternative simple yet systematic route for tuning gene expression, and broadens the present gene regulation toolkit for gene circuit engineering. Beyond the immediate application to circuits in *E. coli*, the synthetic DNA sponge-mediated regulation could also be applied to other prokaryotic and eukaryotic organisms.

## Results

### Design of synthetic DNA sponges for transcriptional regulators.
Here, we aimed to investigate the tuning capacity of synthetic DNA sponge on gene expression regulation using environment-responsive gene circuits as examples. A typical environmentally responsive circuit comprises three cascaded modules: an input sensing module that recognizes and transduces external signals into intracellular transcriptional signals, an optional internal signal processing module which modulates the transduced transcriptional signals (e.g., an amplifier[31] or inverter[32]), and an output module that executes physiological responses (Fig. 1)[33]. In this study, we designed synthetic DNA sponges to sponge off the regulatory proteins within the input sensing module and/or the signal processing module to modulate the circuits' output gene expression. The sponges consist of either pure DNA binding sites or the cognate promoters of the decoyed proteins. For the input sensing module, sponges were designed to decoy the ligand-responsive receptors, while for the signal processing module, sponges were designed to decoy the transcriptional activators or repressors (Fig. 1). In principle, sponging off receptors would lower the intracellular densities of unoccupied receptors, and therefore be able to tune the circuit's output gene expression and sensing sensitivity[12,15]. Sponging off transcriptional factors within the amplifier or inverter module would directly affect their abilities to amplify or invert input transcriptional signals. In addition, the synthetic sponges would allow to address the issue of protein overexpression, reduce system background expression (i.e., leakage) and improve output induction fold (i.e., dynamic range). If the decoyed proteins tend to be toxic at increased expression levels to the host, synthetic sponges may also mitigate their cellular burden.

To test the tuning function of DNA sponges, we designed and standardized two single-layered and three double-layered gene circuits. These circuits include input sensing modules that respond to anhydrotetracycline (aTc), quorum sensing molecule (3OC6HSL) or mercury (HgCl2), and signal processing modules consisting of a typical TetR-based inverter[34], or a transcriptional amplifier based on the ultrasensitive extracytoplasmic function (ECF) sigma factor ECF11_987[15,35]. All circuits were carried on a low copy plasmid (pSEVA121, 4–7 copies per cell)[36]. All sponges were constructed on a medium copy plasmid (pSB3K3, 10–12 copies per cell)[37], flanked downstream by terminators to prevent transcriptional read through. We first tested and validated the function of the single-layer sponges designed for decoying regulatory proteins only in the sensing (Fig. 2) or signal processing module (Fig. 3) individually in circuits using different input sensing modules mentioned before. Multiple repeats of the same sponge site were assembled with 10–30 bp random spacer sequences between to reduce potential interference among adjacent binding events as well as to lower recombination rate (see Methods and Supplementary Data 1). Further, we built dual-layer DNA sponges comprising two different types of decoy binding sites to sponge off regulatory proteins within both the input sensing and internal signal processing modules (Figs. 4, 5), with a hypothesis that such multi-layer DNA sponges may offer cumulative tuning and mitigation effects compared to individual single-layer sponges on the circuit's performance.

### DNA sponges decoy regulators from circuits' input sensing modules.
We first tested whether synthetic DNA sponges can tune an environment-responsive circuit's output expression and sensing sensitivity by sponging off receptor proteins within its input sensing module. Accordingly, two typical small molecule responsive single-layered circuits were chosen for the tests: an aTc-responsive circuit (J23117-30*tetR*-t-P*tet2*-30*gfp*-t) and an

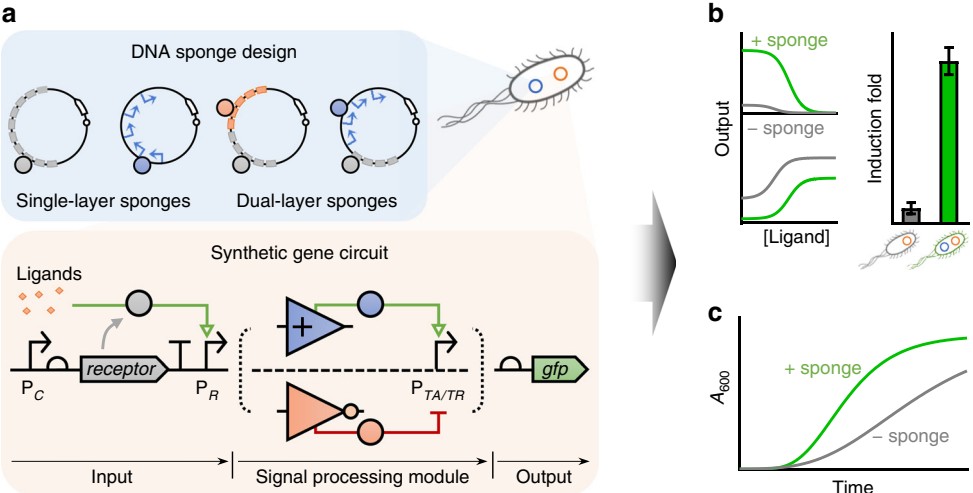

**Fig. 1 Schematic showing DNA sponge as a ubiquitous tool to tune gene expression and mitigate protein toxicity for gene circuit engineering. a** Design of DNA sponges (in blue, top) to tune the response of a target gene circuit (in orange, bottom). A synthetic gene circuit typically comprises an input sensing module, an optional signal processing module (e.g., a transcriptional amplifier or inverter) and an output module for initiating physiological responses. Here in the input sensing module, a constitutive promoter (P$_C$) drives the expression of a receptor gene that responds to target ligands and regulates its cognate promoter P$_R$. In a typical signal processing module, the P$_R$ drives the expression of a transcriptional activator (TA, blue solid circle) or repressor (TR, orange solid circle), which then controls its cognate promoter P$_{TA/TR}$ to express an output protein (e.g., GFP). The DNA sponge is designed to harbor one or multiple protein binding sites (rectangles) or cognate promoters (arrows) of the receptor or TA/TR in the target circuit. **b**, **c** Diagrams showing the effects of the sponge regulation on the circuit's output response (**b**) and growth burden on the host (**c**). With the presence of different DNA sponges (green line/bar) for the receptor, TA or TR, the circuit's basal output expression can be reduced, leading to increased induction fold. In addition, the sponge can absorb excess toxic transcriptional regulators, leading to improved host cell growth.

AHL-responsive circuit (J23101-30*luxR*-t-P$_{lux2}$-30*gfp*-t) (Fig. 2, Supplementary Figs. 3–6).

The aTc-responsive circuit has a constitutive promoter J23117 that drives the expression of the aTc receptor TetR, which would de-repress its cognate promoter P$_{tet2}$ upon aTc binding and trigger the expression of a reporter gene, *gfp*[12] (Fig. 2a). Two types of DNA sponges were designed to decoy TetR: the pure tet operator (tetO)-based sponge excluding sigma factor binding elements and the intact P$_{tet2}$-based sponge which contains two tetO sites. Sponges comprising 1 to 320 tetO repeats were then constructed while sponges comprising only up to 40 repeats of P$_{tet2}$ were successfully assembled due to cloning difficulties (Fig. 2a). In principle, by shunting TetR's binding off its cognate promoter (i.e., the P$_{tet2}$ that drives *gfp* expression) to the tetO/P$_{tet2}$ sponge, the aTc-responsive circuit's output dose-response would be altered. Figure 2b−c shows that the circuit's output leakage (without aTc induction) indeed increased with the increasing number of arrays of the DNA sponges used due to increasingly less output P$_{tet2}$ promoter repressed by TetR, leading to lower induction fold (Fig. 2d). Interestingly, though with the same number of tetO sites, the tetO-containing sponges exhibited bigger impact on the circuit's output response than the P$_{tet2}$-containing sponges (Fig. 2b–d, Supplementary Fig. 4), suggesting the tetO-containing sponge is more efficient than the P$_{tet2}$-containing sponge.

We fitted the circuit's dose-responses to a Hill function-based biochemical model to describe their input-output relationships (see Methods and Supplementary Data 2)[12]. In this case, the Hill constant $K_M$ stands for the inducer concentration provoking half-maximal activation, and is negatively correlated with the system's sensitivity[15]. Here, $K_M$ displayed a clear decline from 0 to 40 repeats of the tetO/P$_{tet2}$ sponge (Fig. 2e), showing a positive effect of the DNA sponge on the repressor receptor-based circuit's sensitivity.

We next continued to test the effect of sponge tuning on the AHL-responsive single-layered circuit. This circuit has a constitutive promoter J23101 that drives the expression of the AHL (3OC$_6$HSL) receptor LuxR, which would activate its cognate promoter P$_{lux2}$ upon 3OC$_6$HSL binding and trigger the output *gfp* expression[12] (Fig. 2f). Here, only the sponges comprising of 1 to 80 repeats of the LuxR binding sites (LBS) were assembled and tested based on the performance of tetO/P$_{tet2}$ sponges tested above. The circuit's dose-responses were fitted to the same Hill function-based biochemical model as aforementioned (Fig. 2g)[12] and the Hill constant $K_M$ was obtained. We observed a horizontal shifting of the circuit's dose-response curves with the presence of an increasing number (1–40) of the LBS-containing sponge arrays (Fig. 2g), resulting in a maximum 10-fold increase of the $K_M$ (Fig. 2h) and therefore a decrease of the activator receptor-based circuit's sensitivity. Such effect was expected and likely caused by a decrease of the intracellular density of the DNA-unoccupied LuxR as the 3OC$_6$HSL-LuxR complex was decoyed away by the LBS-containing sponge[12].

**DNA sponges decoy regulators from signal processing modules.** To further demonstrate the tuning capability of DNA sponge, we proceeded to test the impact of synthetic DNA sponges on decoying different regulatory proteins within the signal processing modules of gene circuits. To this end, we tested their regulatory effects on circuits containing the widely used TetR-based inverter[38] or a recently developed ECF sigma factor-based transcriptional amplifier[15] as its signal processing module (Fig. 3, Supplementary Figs. 7–10).

We first introduced the TetR-based inverter into the aforementioned AHL-responsive circuit, and characterized the dual-layered circuit's response in the presence of the same tetO-containing DNA sponges as described above (Fig. 3a–d). The whole circuit comprises a J23101-*luxR*-P$_{lux2}$ based input sensing module, an internal TetR-based inverter and a final output reporter module (Fig. 3a). We found both circuit's output amplitude (around 3,000 a.u.) and induction fold (around 10 at 1.6 μM 3OC$_6$HSL induction) were low due to the inherent leaky

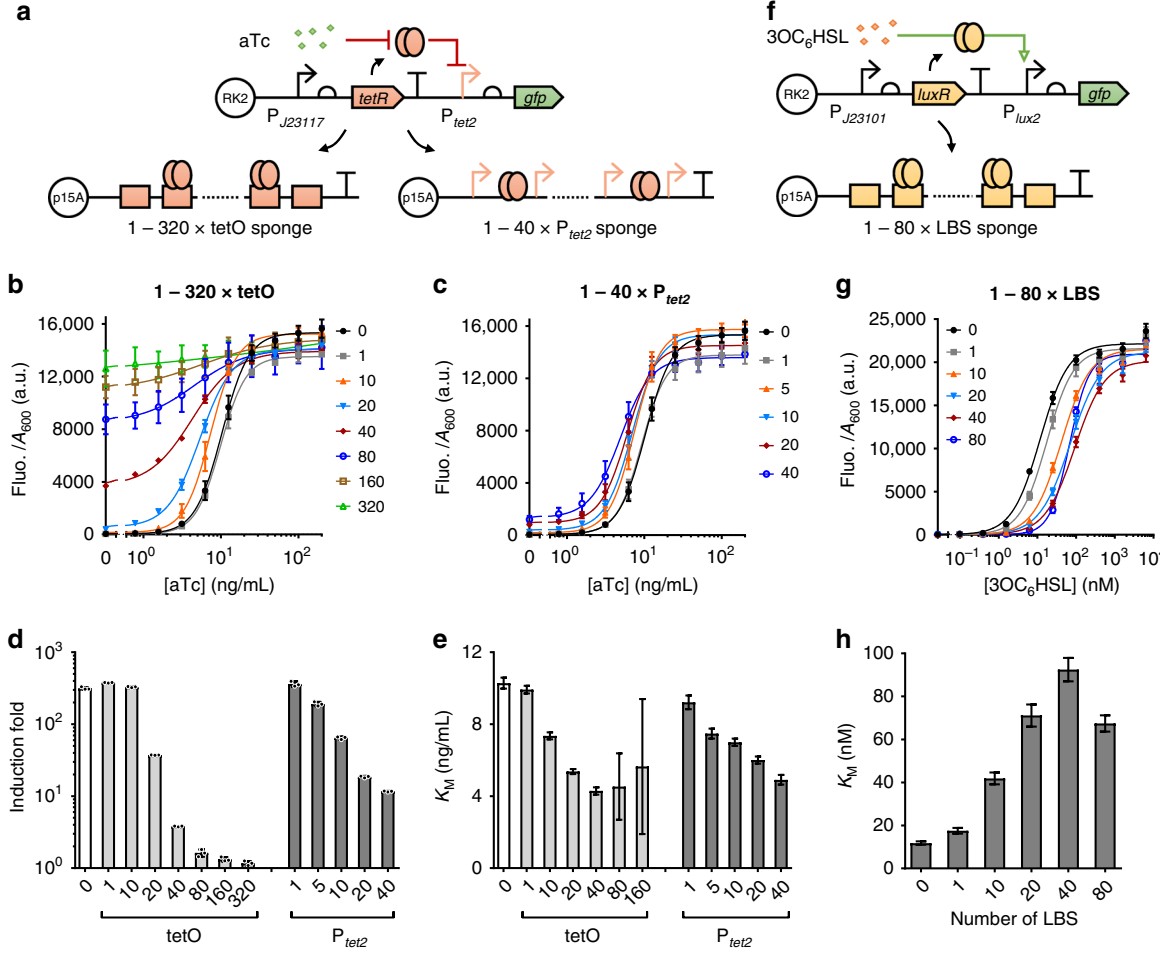

**Fig. 2 Synthetic DNA sponge enables tuning output gene expression by decoying receptors within circuits' input sensing modules. a** Schematic showing the design of two types of synthetic DNA sponges to decoy TetR in an aTc-responsive circuit: 1–320 repeats of TetR operator (tetO) or 1–40 repeats of $P_{tet2}$ promoter. **b, c** The aTc responsive circuit's dose-responses with sponges containing different numbers of the tetO (**b**) or the $P_{tet2}$ repeats (**c**). **d** Induction fold between uninduced and 200 ng/mL aTc induced samples of the circuit with or without the tetO/$P_{tet2}$-based sponge regulation as characterized in (**b**, **c**). **e** Hill constant ($K_M$) of the circuit's fitted dose-responses from (**b**, **c**) against different repeat numbers of the tetO/$P_{tet2}$-containing sponge. $K_M$ value of sample with sponge containing 320 tetO repeats is not shown due to insufficient dose-response of the circuit. **f** Design of an AHL-responsive circuit with regulation by sponges containing 1–80 repeats of the LuxR binding site (LBS). **g** Dose-responses of the AHL-responsive circuit with sponges containing different numbers of LBS repeats. **h** Hill constant ($K_M$) of the circuit's fitted dose-responses from (**g**) against different repeat numbers of the LBS-containing sponge. For (**b**–**d**, **g**), values are mean ± s.d. ($n = 3$ biologically independent samples). For (**e**, **h**), values are mean ± s.e.m. ($n = 3$ biologically independent samples). Fluo., fluorescence. a.u., arbitrary units. Source data are provided as a Source Data file.

expression of TetR even with the use of a very weak ribosome binding site (RBS) B0033[32]. Nevertheless, with the presence of sponges containing 10–80 repeats of the tetO site, the circuit's output amplitude was doubled and the induction fold was increased by 3-fold, demonstrating the capability of sponge for reducing the leakage expression amplified by the internal inverter (Fig. 3b, c). However, using sponges containing excessive tetO sites turned out to reduce the output induction fold due to too much TetR were decoyed, leading to insufficient DNA-unoccupied TetR to repress the output promoter upon induction (Fig. 3b, c). We also tested the impact of the $P_{tet2}$-based sponges on the same circuit. The results obtained reconfirm their inferior decoying ability to the tetO-based sponges as observed before (Supplementary Figs. 4, 7a–c, 8).

The circuit's dose-responses were fitted to the same aforementioned Hill function-based biochemical model (Fig. 3b, d, Supplementary Fig. 7b,d)[32]. The Hill constant $K_M$ and the maximum output $k$ obtained show that the use of sponges increased circuit's output amplitude by 2 to 3-fold though reducing the sensitivity by 10 to 100-fold.

To verify the generality of sponge-mediated reduction on internal expression leakage, we tested it with another two-layer sensing circuit, which contains an amplifier instead of an inverter as its signal processing module. The circuit has a J23109-*merR*-$P_{merT}$ based mercury-responsive input sensing module connected downstream with an ECF11-based transcriptional amplifier[15] (Fig. 3e). As an ultrasensitive high-gain amplifier, it would also amplify the expression leakage from the input sensing module, and was shown to be toxic to the host when the ECF11 activator was at increased expression level[15]. We characterized the two-layered mercury-responsive circuit using sponges containing 1–40 repeats of the $P_{ecf11}$ promoter (Fig. 3e–k). Figure 3f–g confirms that the use of sponges containing 10 and 20 $P_{ecf11}$ repeats reduced the circuit's basal expression and increased its output induction fold, while the induction fold was decreased for sponge containing 40 $P_{ecf11}$ repeats owing to its excessive inhibitory effect on the circuit's output.

Strikingly, we observed significant improvement of host cell growth (up to 25% increase of cell density under 1 µM $HgCl_2$ input induction) when the amplifier circuit was regulated by the

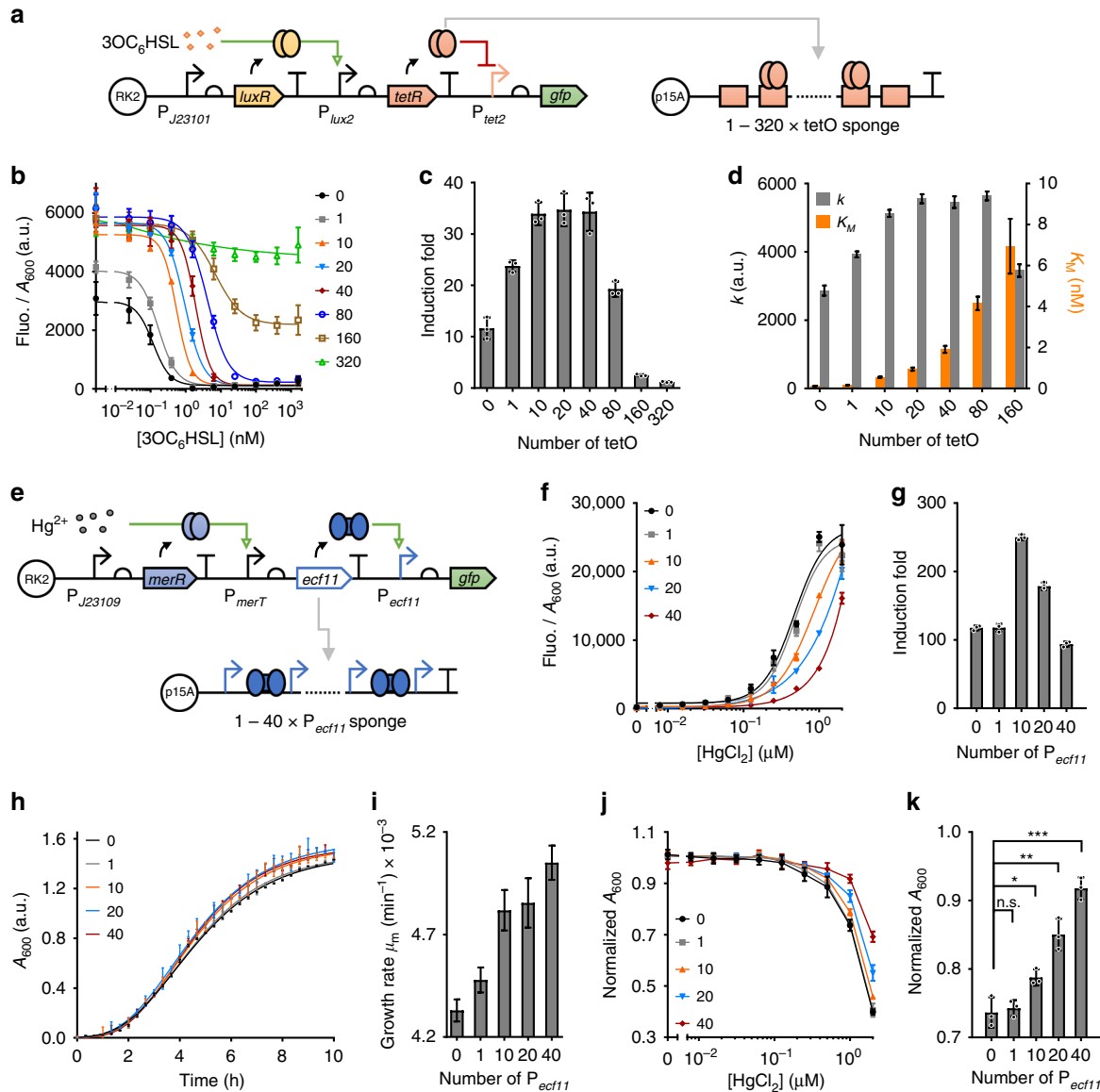

**Fig. 3 Synthetic DNA sponge enables tuning output gene expression and mitigating cellular burden by decoying transcription factors within circuits' signal processing modules. a** Schematic of a two-layered AHL-responsive circuit regulated by sponges containing 1–320 tetO repeats. **b** Characterization of the two-layered sensing circuit in response to various concentrations of 3OC$_6$HSL and with sponges containing 1–320 tetO repeats. **c** Induction fold between uninduced and 1.6 μM 3OC$_6$HSL induced samples of the circuit characterized in (**b**). **d** Hill constant ($K_M$) and the maximum output ($k$) of the circuit's dose-responses characterized in (**b**). Values of sample with sponge containing 320 tetO repeats is not shown due to insufficient dose-response of the circuit. **e** Illustration of a two-layered mercury-responsive circuit regulated by sponge containing 1–40 P$_{ecf11}$ repeats. **f** Dose-responses of the mercury-responsive circuit with sponges containing different numbers of P$_{ecf11}$ repeats. **g** Induction fold between uninduced and 1 μM HgCl$_2$ induced samples of the circuits characterized in (**f**). **h** Gompertz model fitted cell growth curves of the mercury-responsive circuit with sponge containing 1–40 P$_{ecf11}$ repeats in response to 1 μM mercury. **i** Growth rate ($\mu_m$) of the cells at exponential growth phase as characterized in (**h**). **j** Normalized cell densities of the circuits tested in (**f**). Cell densities are normalized to the ones of negative control (with empty sensor and empty sponge plasmids). **k** Normalized cell densities of the circuit induced with 1 μM mercury as characterized in (**j**). Statistical difference is determined by a two-tailed Welch's $t$ test: 1, $p = 0.6987$, t = 0.4257; 10, $p = 0.0381$, t = 3.518; 20, $p = 0.0035$, t = 6.158; 40, $p = 0.0006$, t = 11.44. For (**b**, **c**, **f**–**h**, **j**, **k**), values are mean ± s.d. ($n = 3$ biologically independent samples). For (**d**, **i**), values are mean ± s.e.m. ($n = 3$ biologically independent samples). Fluo., fluorescence. a.u., arbitrary units. $p$ value summary: \*\*\*\*$p$ value < 0.0001, 0.0001 < \*\*\*$p$ value < 0.001, 0.001 < \*\*$p$ value <0.01, 0.01 < \*$p$ value < 0.05, $p$ > 0.05: n.s. Source data are provided as a Source Data file.

P$_{ecf11}$-containing sponges (Fig. 3j, k). This indicates that the circuit's cellular burden was largely caused by the overexpression of ECF11 itself rather than the limitation of cellular resources. To obtain the exact cell growth rate, we fitted the growth curves of the various circuits (under 1 μM mercury induction) to a revised Gompertz model[37] (Fig. 3h). The growth rate ($\mu_m$) obtained was plotted in Fig. 3i, and showed an increase with the increasing

number of arrays of the P$_{ecf11}$-containing sponge used in the circuit.

**Multi-layer DNA sponges enable additive tuning for cascaded circuits.** Built on the results obtained above, we next sought to verify whether multiple-layer DNA sponges can be designed to simultaneously sponge off different regulatory proteins within

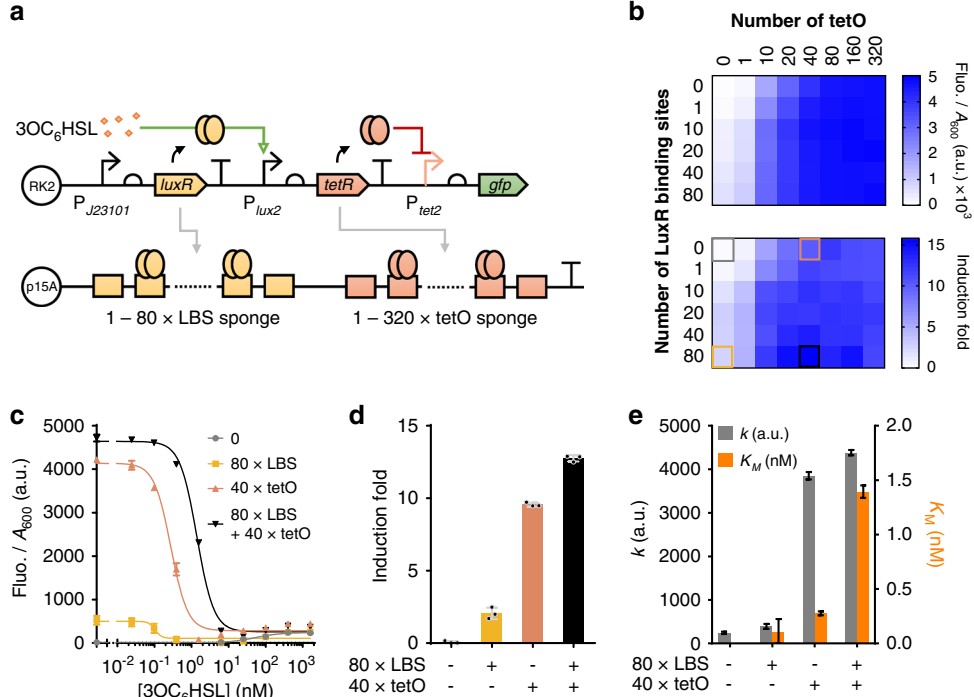

**Fig. 4 Tuning response of a layered circuit using multi-layer DNA sponges to decoy regulators within both the input sensing and signal processing modules. a** Schematic of using dual-layer DNA sponges to tune the response of a layered circuit consisting of an AHL-responsive input sensor connected with a downstream TetR-based inverter. Sponges containing 1–80 repeats of the LuxR binding site (LBS) and 1–320 repeats of the TetR operator (tetO) are combined to decoy LuxR and TetR within the input sensing and signal processing modules respectively. **b** Output gene expression and induction fold of the AHL-responsive circuit with regulation by the LBS-tetO dual-layer sponges. Output gene expression (top panel) is obtained when there is no AHL induction, indicating leaky expression of TetR repressor (the higher the output, the lower the TetR leakage). Induction fold (bottom panel) is calculated using the output fluorescence intensity of the none-induced samples (top graph) divided by the fluorescence intensity of samples induced by 1.6 μM 3OC$_6$HSL (Supplementary Fig. 11a). **c** Dose-responses of the two-layered AHL-responsive circuit with regulation by four different LBS-tetO sponges selected from (**b**). **d** Induction fold between uninduced and samples induced by 1.6 μM 3OC$_6$HSL of the circuit characterized in (**c**). **e** Hill constant ($K_M$) and maximum output ($k$) of the circuit's dose-responses characterized in (**c**). $K_M$ value of sample ($-/-$) is not shown due to insufficient dose-response of the circuit. For (**b**), values are mean ($n = 3$ biologically independent samples). For (**c, d**), values are mean ± s.d. ($n = 3$ biologically independent samples). For (**e**), values are mean ± s.e.m. ($n = 3$ biologically independent samples). Fluo., fluorescence. a.u., arbitrary units. Source data are provided as a Source Data file.

multiple component modules of a circuit, and to provide additive tuning and mitigation effect on the circuit's response and burden on the host. Accordingly, we designed dual-layer sponges to decoy proteins within both the input sensing and internal signal processing modules of exemplar circuits, consisting of an AHL-responsive input sensor connected with a downstream TetR-based inverter or the ECF11-based amplifier (Figs. 4, 5, Supplementary Figs. 11, 12).

We first tested the circuit comprising the TetR-based inverter by combining up to 80 repeats of LBS-containing sponges and up to 320 repeats of tetO-containing sponges to generate the LBS-tetO dual-layer sponges (Fig. 4a). In total 48 single-layer and dual-layer sponges were constructed and characterized to compare their regulatory effects on the circuit's output leakage and induction fold. The results show that the DNA sponges clearly reduced the circuit's leakage and increased its output induction fold, and the tetO-containing single-layer sponge was superior to the LBS-containing single-layer sponges (Fig. 4b). Notably, the dual-layer sponges worked the best and showed an additive tuning effect for both improving the output amplitude (increase up to 132-fold) and induction fold (increase up to 75-fold).

Figure 4c shows the dose-responses of the circuit containing a selected dual-layer sponge (80LBS + 40tetO sponge, black square in Fig. 4b bottom panel) and of those containing none and the

individual single-layer sponges. The results confirm the advantage of using dual-layer sponge over single-layer ones for improving circuit's output amplitude and induction fold (Fig. 4c, d). The model fitted Hill constant $K_M$ and the maximum output $k$ further indicated the cumulative tuning effect of the dual-layer sponge (Fig. 4e).

We next constructed the LBS-P$_{ecf11}$ dual-layer sponges to test their regulatory effect on the two-layered circuit containing the ECF11-based amplifier (Fig. 5a). In total 30 single-layer and dual-layer sponges were assembled and characterized for their impact on the circuit's output leakage, induction fold and host cell growth (Fig. 5b). It shows that the use of sponges resulted in a dramatic reduction of the output basal expression (by 23-fold), a notable increase of the induction fold (by 10-fold, upon 25 nM AHL induction) and a substantial improvement of host cell growth (by 21% of cell density). We next compared the dose-responses, cell densities and growth rates of the circuit containing a selected dual-layer sponge (20LBS + 20P$_{ecf11}$ sponge, black square in Fig. 5b bottom panel) with those of the circuits containing none and the individual single-layer sponges. The results confirmed the ability of DNA sponges to reshape the circuit's response profile and to mitigate the cellular burden arisen from the increased expression of sensitive regulatory proteins in the system (Fig. 5c–i). It is worth noting that the output expression was associated with the cell growth but was not

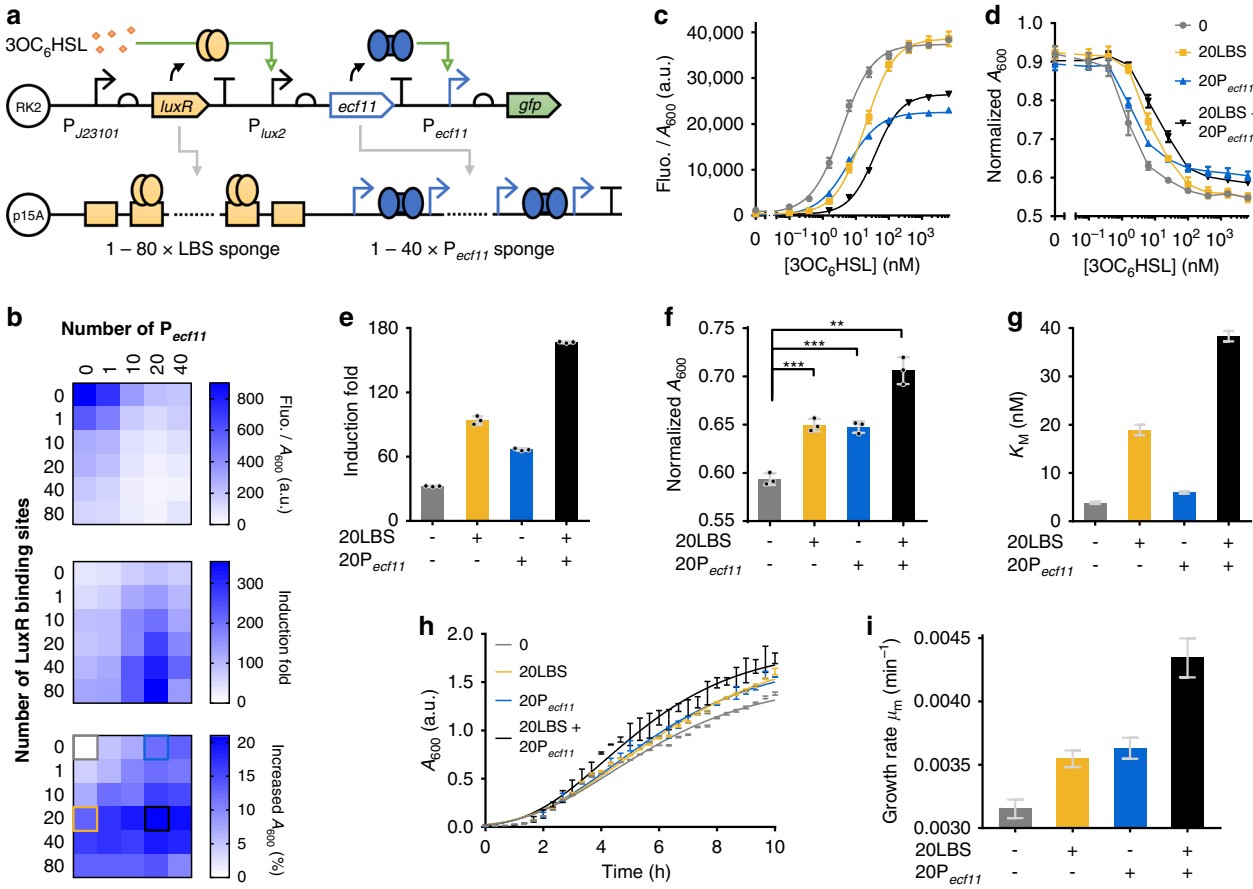

**Fig. 5 Dual-layer DNA sponges provide additive tuning and mitigation effects on a multi-layered circuit's response and burden on the host. a** Schematic of using dual-layer sponges to tune the response of a layered circuit consisting of an AHL-responsive input sensor connected with a downstream ECF11-based amplifier. Sponges containing 1–80 repeats of the LuxR binding site (LBS) and 1–40 repeats of the $P_{ecf11}$ are combined to decoy LuxR and ECF11 within the input sensing and signal processing modules respectively. **b** Output basal expression (top panel, no induction), induction fold (middle panel) and improved cell density (bottom panel) of the circuit with LBS-$P_{ecf11}$ dual-layer sponge regulation. Induction fold is calculated using the circuit's output fluorescence upon 25 nM 3OC$_6$HSL induction (Supplementary Fig. 12a) divided by the basal output expression. Increased percentage in cell density is calculated as: ($A_{600}$(sample with sponge)/$A_{600}$(sample without sponge) − 1) × 100. **c, d** Dose-responses (**c**) and normalized cell densities (**d**) of the circuit regulated by the sponges selected from (**b**). Cell densities are normalized to the negative control (carrying empty circuit and empty sponge plasmids). **e** Induction fold between uninduced and 6.4 μM 3OC$_6$HSL induced samples of the cell strains characterized in (**c**). **f** Normalized cell densities of the cell strains induced with 25 nM 3OC$_6$HSL as tested in (**c**). Statistical difference is determined by a two-tailed Welch's t test: (+/−), p = 0.0004, t = 10.86; (−/+), p = 0.0004, t = 10.91; (+/+), p = 0.0016, t = 12.78. **g** Hill constant ($K_M$) of the fitted dose-responses of the circuit characterized in (**c**). **h** Gompertz model fitted growth curves of the cell strains from (**c**) in response to 25 nM 3OC$_6$HSL. **i** Growth rate ($\mu_m$) of the cell strains at exponential growth phase as characterized in (**h**). For (**b**), values are mean (n = 3 biologically independent samples). For (**c-f, h**), values are mean ± s.d. (n = 3 biologically independent samples). For (**g, i**), values are mean ± s.e.m. (n = 3 biologically independent samples). Fluo., fluorescence. a.u., arbitrary units. p value summary: ****p value < 0.0001, 0.0001 < ***p value < 0.001, 0.001 < **p value < 0.01, 0.01 < *p value <0.05, p > 0.05: n.s. Source data are provided as a Source Data file.

the major cause of the reduced cell densities observed, since another genetic circuit with similar output level was not shown to be toxic to the host cells (Fig. 2g, Supplementary Fig. 3j), supporting that the cellular burden was largely caused by the expression of ECF11 itself. Due to the $P_{ecf11}$ sponges decoy ECF11 without affecting its expression level while LBS sponges indirectly decrease ECF11 expression, our results suggest that the metabolic burden results from the high expression level of ECF11 and its binding activity. This is in line with a previous study which suggested that the toxicity of ECF could derive from RNA polymerase (RNAP) competition for native RNAP with host sigma factors and/or from aberrant gene expression[35]. The model fitted Hill constant $K_M$, maximum output $k$, and growth rate $\mu_m$ further show the additive tuning effect of the dual-layer sponge (Fig. 5g, i), and the advantage of using dual-layer sponge over single-layer ones for improving circuit's performance (Fig. 5e–i).

**Genetic stability and noise effect of synthetic DNA sponges.** To support effective and long term use, synthetic DNA sponges should allow stable inheritance in their host cells. Given the large number of repetitive genetic elements present in the sponges, we proceeded to test the genetic stability of the sponges in their hosts. Although the *E. coli* TOP10 strain we used for sponge regulation study is *recA* deficient, there may be RecA-independent recombination occurring in the cells. To this end, we selected the largest sponge from each single-layer sponge type (i.e., 320 × tetO, 40 × $P_{tet2}$, 80 × LBS, 40 × $P_{ecf11}$) and two dual-layer sponges (i.e., 80 × LBS + 40 × tetO and 20 × LBS + 20 × $P_{ecf11}$) to test their stability across 100 generations (5-day serial dilution) in their hosts (see Methods, Fig. 6a, Supplementary Fig. 13). We determined the plasmids stability by analyzing DNA fragment sizes after restriction digestion, using gel electrophoresis. Figure 6a shows that most sponge plasmids are stable over 100 generations except

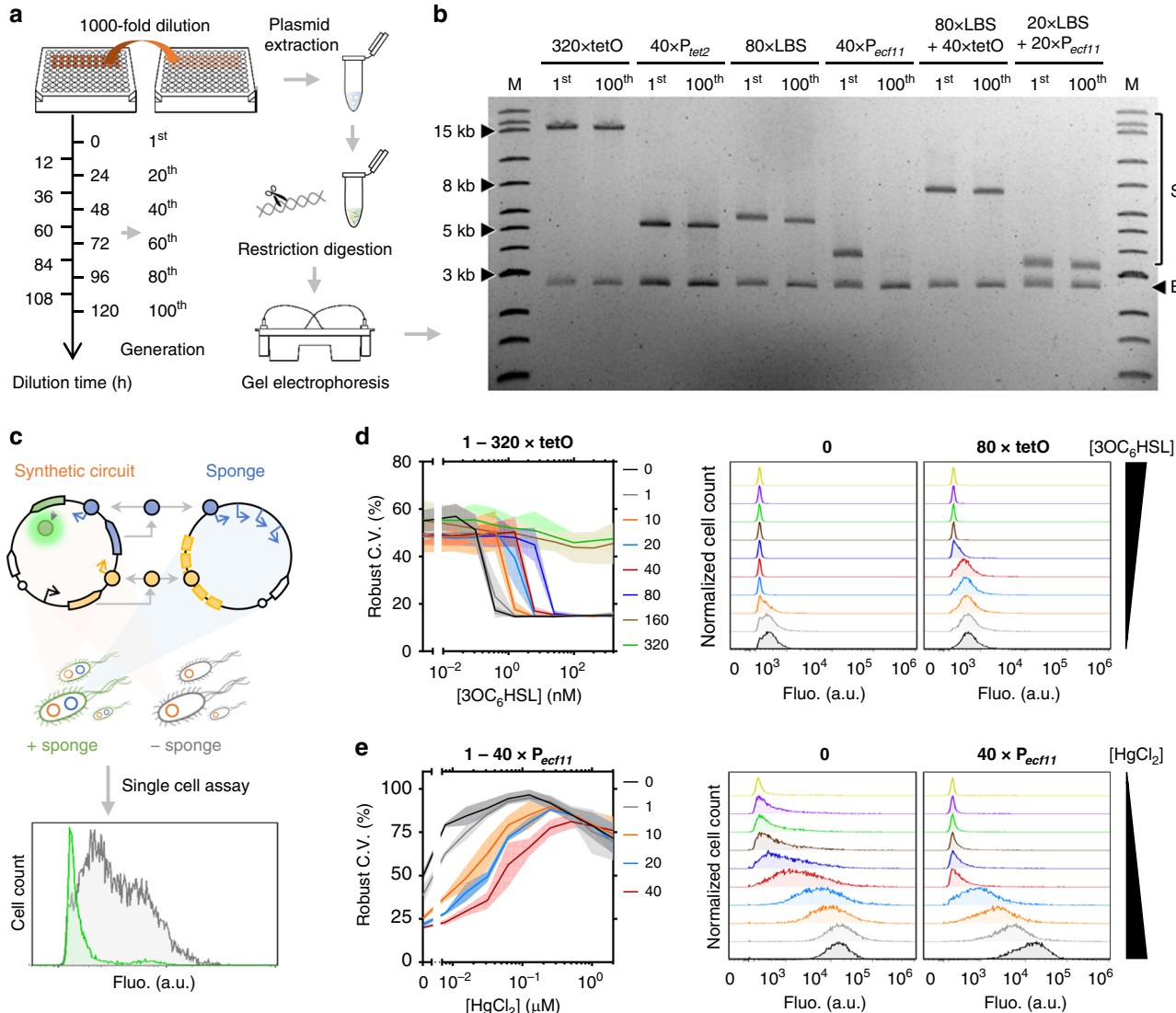

**Fig. 6 Genetic stability of synthetic DNA sponge and its effect on circuit's output gene expression noise. a** Schematics showing the workflow of the stability assay of synthetic DNA sponges. The cells harboring the sponge plasmids were cultured in 1 mL of medium in a 96-deep well plate and were diluted every 12 h for 5 days (corresponding to approximately 100 generations in total). Every 24 h, the cells were harvested for plasmid extraction, restriction digestion and the plasmids were analyzed by electrophoresis (see Methods). **b** Image of a gel post electrophoresis showing the stability of synthetic single-layer DNA sponges of 320 × tetO, 40 × P$_{tet2}$, 80 × LBS, 40 × P$_{ecf11}$ and dual-layer DNA sponges of 80 × LBS + 40 × tetO and 20 × LBS + 20 × P$_{ecf11}$. The sponge plasmids extracted from the host cells after 1 or 100 generations were compared. The plasmids from 20th, 40th, 60th and 80th generations and another two biological replicates showing similar results are shown in Supplementary Fig. 13. The expected sizes of the selected DNA sponges are shown in Table 1. M, DNA marker. S, sponge. B, pSB3K3 backbone. The original image is provided as a Source Data file. **c** Schematics showing the study of noise effect of synthetic DNA sponge on synthetic circuit's output gene expression. The output fluorescence was compared at single cell level between the host cells with and without the presence of sponge. The noise was measured on the basis of robust coefficient of variation (C.V.) of the circuit's output gene expression. **d** Robust C.V. of the output gene expression (left) and dose-responses (right) of a two-layered AHL-responsive circuit (Fig. 3a) with or without tetO sponges from single cell assays. Values are mean ± s.d. indicated by shading (n = 3 biologically independent samples). The cells were induced with 0, 0.02, 0.10, 0.39, 1.56, 6.25, 25, 100, 400 and 1,600 nM of 3OC$_6$HSL. **e** Robust C.V. of the output gene expression (left) and dose-responses (right) of a two-layered mercury-responsive circuit (Fig. 3e) with or without P$_{ecf11}$ sponges from single cell assays. Values are mean ± s.d. indicated by shading (n = 3 biologically independent samples). The cells were induced with 0, 0.008, 0.016, 0.031, 0.063, 0.125, 0.25, 0.5, 1 and 2 μM of HgCl$_2$. Full profile of dose-responses at single cell level and noise effect of other sponges are shown in Supplementary Figs. 5, 6, 9, 10g, 11d, 12i. Fluo., fluorescence. a.u., arbitrary units. Source data are provided as a Source Data file.

the 40 × P$_{ecf11}$ sponge which has been gradually lost in its host cells (Supplementary Fig. 13). This suggests that the synthetic DNA sponges are generally stable and that multi-layer sponges with a lower number of repeats for each individual sponge may perform better in terms of genetic stability, when compared to a single-layer sponge with a high number of sponge repeats.

One challenge in synthetic circuit design is to minimise their effect on gene expression noise introduced and enhance the robustness of cellular response. In principle, decoying transcriptional factors may affect such noise significantly as it directly interferes with gene expression at transcriptional level, which has been shown to be the dominant source of gene expression noise[39]. To investigate this, we evaluated how synthetic DNA sponges

affected the noise of the output gene expression in our designed circuits (Fig. 6c, d). The noise was measured on the basis of robust coefficient of variation (C.V.) of the circuits' output gene expression at single cell level. We found that noise was generally increased when DNA sponges were present for sponging off transcriptional repressors (i.e., TetR) (Fig. 6d, Supplementary Figs. 5, 9, 11d). For sponging off transcriptional activators (i.e., LuxR and ECF11), such effect turned out to be the opposite when low level activators were available for sequestration (Fig. 6e, Supplementary Fig. 10g), but remained for high activator levels in some cases (Supplementary Figs. 6, 12i). This suggests that the effect of protein decoying on gene expression noise may differ on a case-by-case basis and depend on the type and ratio of the decoyed proteins and decoys available.

## Discussion

We showed for the first time that synthetic DNA sponge-mediated protein sequestration can systematically regulate gene expression within gene circuits with multifaceted quantitative tuning capacities and mitigate the burden of protein expression on host cells. This approach provides a simple yet generalizable route for addressing many commonly met issues in gene circuit design such as intolerable basal expression leakage, low output amplitude and narrow dynamic range, and cellular burden induced by protein expression toxicity[7,15,37,40–43]. Using environmentally responsive multi-layered circuits as examples, we showed that both the input sensing and signal processing modules can be regulated separately by individual single-type DNA sponges, whereas sponging both modules simultaneously leads to an additive tuning effect on circuits' performance. It is worth noting that only limited numbers of repeats of some single-type sponges (e.g., LBS, $P_{tet2}$ and $P_{ecf11}$ sponges) have been constructed on the sponge plasmid due to difficulties met in cloning, perhaps arising from increased instability for large numbers of those DNA repeats, while multi-layer sponges may offer an alternative solution to bypass this limitation. Importantly, unlike many typical transcriptional and translational regulation methods which may increase cellular burden due to the need of expression of additional protein components, the DNA sponges used did not show any notable growth burden to the host cells (Supplementary Figs. 3e,f,j, 7h, 11c). On the contrary, they were shown to benefit host growth in certain cases by sponging away sensitive transcription factors that tend to be burdensome at increased expression levels (Fig. 3h–k, 5). This indicates that synthetic DNA sponge could be repurposed as a handy tool for identifying toxic components in gene circuits. Further, by varying the number of arrays and types of the DNA sponges used, the circuits' performance could be precisely tuned with bespoke response, which is important for cascading multi-layered gene circuits to achieve complex cellular signal processing.

Consistent with some previously reported theoretical studies[44–46], our study showed that the effect of protein decoying on gene expression noise depended on the type and expression level of the decoyed transcription factors as well as the ratio of the decoyed proteins and decoys available. Notably, the increase in noise did not result from DNA recombination because our synthetic DNA sponges were shown to be generally stable in their host cells.

Interestingly, we noted that the DNA sponges with TetR binding sites alone (i.e., tetO sponge) exhibited better sponge tuning effect than the $P_{tet2}$ promoter-based sponges (i.e., $P_{tet2}$ sponge). We view this may be due to the interference of transcription initiation on the $P_{tet2}$ promoter-based sponges, leading to their lower binding affinity compared to the sponges containing pure TetR binding sites. The observation is consistent with a prior study which used chromatin immunoprecipitation (ChIP) assay to show that the tet-transcriptional-activator (tTA)

binding to decoys comprising pure tTA binding sites was stronger than that to its target promoters in yeast[23]. Nevertheless, our approach provides a simpler and cost effective way to differentiate the binding affinities of these alternative sponge designs.

Surprisingly we found that the presence of empty plasmids alone could already affect the circuits' output response and host growth in most cases (Supplementary Figs. 3b–d,h,i, 7e–g, 10a–f, 11b, 12b–h), though such tuning effect became more drastic with the additional presence of synthetic DNA sponges on the plasmids. We view this could be due to the competition and reallocation of host resources induced by the presence of the empty plasmids. Since a portion of the total cellular resources will be consumed by the backbone plasmids, less would be available for use by the circuits and host endogenous networks, leading to the observed alterations on circuit response and host growth.

In summary, our work shows that synthetic DNA sponge provides a generalizable tool for systematically tuning gene expression within synthetic gene circuits. Moreover, alternative DNA sponges with mutant sequences of varying binding affinities to the same decoyed protein can be designed to further broaden the tuning space for gene circuit engineering. The noise effect would be of interest for further investigation since our results did not show a conclusive unifying effect of DNA sponge on buffering noise of target gene expression. In addition, our study suggests that synthetic DNA sponge may facilitate other fundamental research areas such as investigating the binding affinities of proteins to their cognate or non-cognate DNA binding sites on the genome. In contrast to traditional methods such as ChIP assay[47], an assay developed utilizing synthetic DNA sponges would be much simpler especially for initial screening of different candidate binding sites. Lastly, synthetic DNA sponge has potential to act as a handy tool to pinpoint potentially toxic components within a synthetic circuit and subsequently to mitigate their metabolic burden placed on the host.

## Methods

**Strains, plasmids and growth conditions**. Plasmid cloning and gene circuit characterization were all performed in *E. coli* TOP10. Cells were cultured in lysogeny broth (LB) medium (10 g L$^{-1}$ peptone (EMD Millipore), 5 g L$^{-1}$ NaCl (Fisher Scientific), 5 g L$^{-1}$ yeast extract (EMD Millipore)), with appropriate antibiotics. The antibiotic concentrations used were 50 µg mL$^{-1}$ for both kanamycin and ampicillin (for low copy number plasmid), or 100 µg mL$^{-1}$ for ampicillin (for high copy number plasmid).

For circuit characterization, a single colony of fresh transformant of engineered bacteria from solid LB plate was first suspended into 50 µL fresh LB medium with appropriate antibiotics. 5 µL of the suspension was inoculated into 195 µL fresh LB medium with appropriate antibiotics in a flat-bottom 96-well CytoOne® Plate (SLCC7672-7596, Starlab), which was then incubated in a shaker incubator (MB100-4A, Allsheng) with continuous shaking (1,000 rpm, 37 °C) for 17 – 18 h overnight. Thereafter, 2 µL of the overnight culture was added into 193 µL fresh LB medium with appropriate antibiotics in a black 96-well microplate with clear bottom (655096, Greiner Bio-One) and induced with 5 µL inducers to a final volume of 200 µL per well. The microplate was sealed with an air permeable film (AXY2006, SLS), and incubated in the same shaker incubator with continuous shaking (1,000 rpm, 37 °C). After 5 h unless otherwise indicated, a plate reader (BMG FLUOstar) was used to measure the fluorescence and absorbance of the cell culture.

Antibiotics, and inducers mercury (II) chloride (HgCl$_2$) and N-(β-ketocaproyl)-L-homoserine lactone (3OC$_6$HSL) were analytical grade and purchased from Sigma-Aldrich. Inducer anhydrotetracycline (aTc) was analytical grade and was purchased from Cayman Chemical. Ampicillin, kanamycin, HgCl$_2$ and 3OC$_6$HSL were dissolved in ddH$_2$O before filtered using 0.22 µm syringe filters (SLGP033RS, Millipore). aTc was dissolved in 50% ethanol.

**Plasmid circuit construction**. Standard molecular biology techniques were used to construct the plasmids containing the environment-responsive gene circuits and the sponge arrays. Key plasmids used in this study are summarized in Supplementary Data 1, and detailed plasmid maps are provided in related figures and Supplementary Figs. 1, 2. Parts sequences and sources are listed in Supplementary Data 1. All oligonucleotides used in this study are listed in Supplementary Data 1 and were purchased from Sigma-Aldrich and Integrated DNA Technologies. All plasmids constructed in this study have been confirmed by Sanger sequencing (Source BioScience) and restriction digestion.

All environment-responsive gene circuits were on a low copy RK2-origin plasmid (pSEVA121, Genebank: JX560322.1)[48]. All sponge arrays were on a medium copy p15A-origin plasmid (pSB3K3, http://biobricks.org). Qiaspin Miniprep Kit (Qiagen) and Wizard SV Gel and PCR Clean-Up System (Promega) were used for DNA purification.

Each single sponge construct (i.e., 1 × tetO[49]/P$_{tet2}$[12]/LBS (LuxR binding site)[50]/P$_{ecf11}$[15]) was constructed by annealing oligonucleotides harboring the operator or promoter, which were then ligated with a terminator (BBa_B0015) into a pSB3K3 plasmid backbone by BioBrick assembly[51]. 10 × tetO sponge was designed with 10–30 bp of random neutral DNA sequences between each tetO binding site to reduce the possibility of recombination, and was constructed by annealing five pairs of oligonucleotides with each harboring two tetO binding sites, followed by ligation with BBa_B0015 into a pSB3K3 plasmid backbone by Golden Gate[52] and BioBrick assembly. 10 × tetO without terminator was built in pSB1A3 plasmid for further construction. 20 × tetO sponge was constructed by BioBrick assembly of the 10 × tetO sponges with and without the terminator. Similarly, up to 320 × tetO sponges were built up. 5 × P$_{tet2}$ sponge was constructed by Golden Gate and BioBrick assembly of five PCR amplified P$_{tet2}$ sequences and a BBa_B0015 terminator. Up to 40 × P$_{tet2}$ sponges were built up by BioBrick assembly. 10 and 20 × LBS sponge were built via annealing oligonucleotides, PCR, Golden Gate and BioBrick assembly, and the 20 × LBS sponge was used to construct up to 80 × LBS sponges. Likewise, 2 × P$_{ecf11}$ sponge was first created for the construction of up to 40 × P$_{ecf11}$ sponges. Apart from the tetO sponge, more repeats of the other sponges used could not be generated due to cloning difficulties. Dual sponges were assembled via BioBrick assembly. For example, 10 × LBS sponge without terminator was ligated with 40 × tetO sponge with terminator to construct the 10LBS-40tetO dual-layer sponge.

**Gene expression assays and data analysis**. The growth conditions for characterizing the engineered circuits are described above. The plasmid containing the environment-responsive gene circuits were co-transformed with the plasmid carrying the sponge arrays for testing. The circuits co-transformed with empty pSB3K3 plasmid was defined as the 0 sponge control. Green Fluorescent Protein (GFP) was used as the reporter for all circuits. Its expression level was measured by a plate reader (BMG FLUOstar, bottom reading, with 485 nm excitation and 520 ± 10 nm emission wavelengths, Gain = 1,000). Absorbance (top reading, $A_{600}$) was also read at the same time to determine the cell density.

The fluorometry data were first processed using Omega MARS Data Analysis Software and then exported to Microsoft Excel 2013 and GraphPad Prism v8.1.2 for data analysis. The background fluorescence and absorbance of the medium were determined from blank wells loaded with LB media and were subtracted from the readings of the other wells. The fluorescence/$A_{600}$ (Fluo./$A_{600}$) at 5 h growth post induction and incubation (when cells were in exponential growth and the steady state assumption for protein expression is applied) for a sample culture was determined by subtracting its triplicate-averaged counterpart of the negative control cultures (GFP-free) at the same time. Each sensor was tested with three biologically independent replicates. All the data shown are mean values and were acquired using the plate reader unless otherwise indicated.

Flow cytometry assays were performed following the plate reader assays. Briefly, the cells from the 96-well plate were transferred with 1:100 dilution to another U-bottom 96-well plate (612U96, Fisher Scientific) with PBS (1 ×, with 2 mg mL$^{-1}$ Kanamycin to stop translation), which was then incubated at 4 °C for at least 1 h. 10,000 total cell events were collected at low flow rate using Attune NxT 3.1.2.0 software on an Attune NxT Flow Cytometer (equipped with Attune NxT Autosampler, using 488 nm excitation laser and a 530 nm emission filter with 30 nm bandpass). The flow cytometry date were first processed using Attune NxT 3.1.2.0 software with an appropriate gate of forward-scattering and side-scattering for all tested cultures and then were exported to Microsoft Excel 2013, GraphPad Prism v8.1.2 and FlowJo 7.6.1 for data analysis. Supplementary Fig. 14 exemplifies the gating strategy.

**Mathematical modelling and data fitting**. Biochemical models were developed for individual transcription factor receptor modules to abstract their ligand-dependent dose response behaviors. As described previously, the ordinary differential equation-based deterministic model was used for accurately modelling the gene regulation and expression across the full input or output range of the sensor systems[12,15,32].

The resulting data from Figs. 2b,c,g, 3f, 5c were fitted to a Hill function model for the steady-state input-output response (transfer function) of an inducible promoter in the form[12,32]:

$$f([I]) = \alpha_1 k_1 + k_1 [I]^{n_1}/(K_{M1}^{n_1} + [I]^{n_1}) \quad (1)$$

where [I] is the input inducer concentration, $K_{M1}$ and $n_1$ are the Hill constant and coefficient respectively, relating to the promoter-regulator/inducer interaction, $k_1$ is the maximum expression rate without repression and $\alpha_1$ is a constant relating to the basal activity level of the input promoter due to leakage.

The data from Figs. 3b, 4c were fitted to the transfer function of an inducible repressor-based inverter circuit in the form[32]:

$$f([I]) = \alpha_2 k_2 + k_2 K_{M2}^{n_2}/(K_{M2}^{n_2} + [I]^{n_2}) \quad (2)$$

**Table 1 DNA sponges selected for plasmid stability assay.**

| DNA sponge | Length without plasmid backbone |
|---|---|
| 320 × tetO | 14.9 kb |
| 40 × P$_{tet2}$ | 5.0 kb |
| 80 × LBS | 5.5 kb |
| 40 × P$_{ecf11}$ | 3.7 kb |
| 80 × LBS + 40 × tetO | 7.4 kb |
| 20 × LBS + 20 × P$_{ecf11}$ | 3.3 kb |

where [I] is the input inducer concentration, $K_{M2}$ and $n_2$ are the Hill constant and coefficient, $k_2$ is the maximum expression rate due to induction and $\alpha_2$ is a constant relating to the basal activity level of the input promoter due to leakage.

Gompertz model[37,53] was used to fit the cell growth curves (Figs. 3h, 5h) for each sample with measured cell density ($A_{600}$). The growth curves were defined as the logarithm of the cell density (y) plotted against time (t):

$$y = A \, exp\left\{-exp\left[\frac{\mu_m \, e}{A}(\lambda - t) + 1\right]\right\} \quad (3)$$

where $\mu_m$ stands for the growth rate of the bacteria at exponential growth phase; A is the maximum cell density that could be achieved by the cell culture; $\lambda$ is the lag time before the bacteria entering exponential growth phase.

GraphPad Prism v8.1.2 was used to fit the transfer function to the characterized dose-responses of circuits, and the best fit parameters and their standard errors are listed in Supplementary Data 2.

**Sponge plasmids stability assays**. E. coli TOP10 was transformed with selected sponge plasmids as listed in Table 1, plated on LB agar and grown for 17 h at 37 °C. A single colony of each sponge was inoculated into 1 mL terrific broth (TB, containing 12 g L$^{-1}$ tryptone, 24 g L$^{-1}$ yeast extract, 0.4% (v/v) glycerol, 17 mM KH$_2$PO$_4$, and 72 mM K$_2$HPO$_4$) in a 2.0 mL 96 Deepwell Plate (E2896-2110, StarLab). The liquid culture was grown for 17 h at 37 °C in the MB100-4A shaker incubator (Allsheng) with continuous shaking (1,000 rpm). The resulting cells were marked as their first generation. Then the cells were back diluted 1,000-fold into 1 mL fresh TB every 12 h for 5 days, which corresponded to approximately 10 × log$_2$ 1000 ≈ 100 generations with 10 generations for each transfer. The cells at 1st, 20th, 40th, 60th, 80th and 100th generations were harvested for sponge stability analysis. 50 µg mL$^{-1}$ kanamycin was used for all related experiments. Three colonies of each sponge were tested as three biological replicates.

To confirm the number of cell generations, the cell densities of the cell cultures before each transfer were determined by absorbance ($A_{600}$). 10 µL cell cultures were diluted into 90 µL fresh TB in a black 96-well microplate with clear bottom (655096, Greiner Bio-One), and the $A_{600}$ was measured using the BMG FLUOstar plate reader. The background absorbance of the medium was determined from blank wells loaded with TB media and was subtracted from the readings of the other wells. The absorbance data were first processed using Omega MARS Data Analysis Software and then were exported to Microsoft Excel 2013 for data analysis. For each 12 h period, the number of cell generations were calculated using the following equation:

$$\text{Cell generations} = \log_2(A_t/A_0) \quad (4)$$

where $A_0$ is the $A_{600}$ value of the cell culture following dilution and $A_t$ is the $A_{600}$ value of the diluted cell culture post 12 h incubation.

To analyze the sponge plasmids stability, we purified the sponge plasmids and compared the size of the plasmids from the 20th, 40th, 60th, 80th and 100th generations with their 1st generation. Qiaspin Miniprep Kit (Qiagen) was used to purify the plasmid DNA, which was then quantified by DS-11+ Spectrophotometer (DeNovix). Unless otherwise indicated, 600 – 800 ng plasmid DNA was digested using FastDigest$^{TM}$ restriction enzymes XbaI and PstI (ThermoFisher), and the digested DNA fragments were analyzed by electrophoresis using 0.5% (w/v) TBE (20-6000-50, Severn Biotech) agarose gel (16500500, ThermoFisher). 4 uL Quick-Load® 1 kb Extend DNA Ladder (NEB) was used as DNA size marker. The gel was run at 120 V for 70 min in a Wide Mini-Sub Cell GT Cell (Bio-Rad), followed by post-staining in SYBR™ Safe DNA Gel Stain (S33102, ThermoFisher) diluted 10,000-fold in 200 mL TBE at room temperature for 30 min with 40 rpm shaking incubation (on a ProBlot™ Rocker 25). For the tests shown in Supplementary Fig. 13, 550 – 850 ng of DNA were digested with XbaI and PstI for 90 min at 37 °C before gel electrophoresis. The gels were pre-stained with SYBR™ Safe, and were ran at 400 mA constant current for 40 min. The gels were imaged using a Bio-Rad Gel Doc XR+ system with filter 1 and SYBR Safe mode. The gel images were acquired and adjusted by Image Lab software at 600 dots-per-inch resolution.

**Reporting summary**. Further information on research design is available in the Nature Research Reporting Summary linked to this article.

## Data availability

All data and plasmids supporting the findings are available from the corresponding author upon reasonable request. Plasmids of the six stimuli-responsive genetic circuits, all the single-layer DNA sponges, and representative dual-layer sponges are available from Addgene (ID in brackets): pXW117Ptet2-gfp (160818), pXW101Plux2-gfp (160819), pXW101Plux2-33Ptet2-gfp (160820), pXW101Plux2-32Ptet2-gfp (160821), pXW109PmerT-Pecf11-gfp (160822), pXW101Plux2-Pecf11-gfp (160823), pXWStetO (160824), pXWS10tetO (160825), pXWS20tetO (160826), pXWS40tetO (160827), pXWS80tetO (160828), pXWS160tetO (160829), pXWS320tetO (160830), pXWSPtet2 (160831), pXWS5Ptet2 (160832), pXWS10Ptet2 (160833), pXWS20Ptet2 (160834), pXWS40Ptet2 (160835), pXWSLBS (160836), pXWS10LBS (160837), pXWS20LBS (160838), pXWS40LBS (160839), pXWS80LBS (160840), pXWSPecf11 (160841), pXWS10Pecf11 (160842), pXWS20Pecf11 (160843), pXWS40Pecf11 (160844), PXWS80LBS40tetO (160845), PXWS80LBS320tetO (160846), PXWS20LBS20Pecf11 (160847), PXWS80LBS40Pecf11 (160848). Source data are provided with this paper.

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

## Acknowledgements

This work was supported by a UK Research and Innovation Future Leaders Fellowship [MR/S018875/1], UK Biotechnology and Biological Sciences Research Council grant [BB/N007212/1], Leverhulme Trust research project grant [RPG-2020-241] and Wellcome Trust Seed Awards in Science [202078/Z/16/Z].

## Author contributions

B.W. conceived and led the study. X.W. and B.W. designed the experiments. X.W. performed the experiments and data analysis with input from B.W. and Y.L. X.W., F.P. and Y.L. designed, performed and analyzed the experiments of DNA sponge stability assay. B.W. and X.W. wrote the manuscript with comments from all authors.

## Competing interests

The authors declare no competing interests.
