## [Peer Review File · Nature Communications]

Reviewers' Comments:

Reviewer #1:

Remarks to the Author:

This is an interesting and important collection of work, demonstrating the utility of an elegantly simple method ("DNA sponges") in shaping the response curves of synthetic regulatory circuits. The ability to shape such curves effectively is vital to realizing synthetic biology's path towards being able to design genetic circuits with the kind of flexibility currently enjoyed in other branches of engineering. A substantial number of experimental tests of the system have been presented, making a significant case for the utility of the approach. Beyond the immediate application to circuits in *E. coli*, the general idea has more general relevance in other organisms, which I believe makes it of wide enough interest to appear here rather than in a more specialized journal.

Some comments/questions:

- Would it be possible to clarify further what is happening in Figure 5? The best-growing (and thus least-burdened) condition is +20 LBS / +20 Pecff11, but that condition also produces either the lowest or second-lowest level of GFP reporter output, if I'm interpreting panel (c) correctly. Level of reporter expression correlates with metabolic burden, so is the reduction in burden just a function of that effect? The fold activation is highest under those +20/+20 conditions, so perhaps the idea is to achieve the highest fold activation with the lowest required output level, or something similar? There's a reference in the Discussion to "sponging away sensitive transcription factors that tend to be burdensome at increased expression levels", but it wasn't entirely clear to me in the discussion of Figure 5 where/how this is happening. Some additional discussion, where Figure 5 is first introduced, of the mechanics underlying the improved growth rate (as a measure of decreased metabolic burden) would be helpful.

- I found the structure of the Discussion section a bit unusual: Quite a bit of the section is spent introducing whole new types of results (measurements of the noise in gene expression, for example) that appear only in the Supplementary Information. It seems more appropriate to include those in the Results section.

David McMillen

Reviewer #2:

Remarks to the Author:

In this manuscript, Wan & Wang show a novel approach to use DNA sponges to sequester prokaryotic transcription factors and shift regulatory equilibriums in cells.

Although the data is solid and well presented, the reviewer doubts that the data here has enough potential to be interesting to researchers in other related disciplines, and therefore justify publication in *Nature Communications*. Although, this work is surely of novelty and is relevant for scientists in the specific field, and should therefore be published in a more specialized journal.

To improve the quality of the manuscript, some general comments:

It would be helpful for the reviewer to have line numbering so it would be easier for the reviewer to point to a certain position in the text.

The reviewer was puzzled until the end of the introduction which type of organism this work is referring to. The authors should be clearer from the beginning that they are talking about prokaryotes, and their experiments are performed in *E. coli*. The reader should not be left thinking this work refers to eukaryotic systems until just before the results part begins.

The authors never actually show binding of the targeted transcription factors to their DNA sponges. An experimental setup showing direct, physical interaction such as ChIP should be performed.

Abstract:

What do the authors mean by "hidden gene regulation mechanism from nature". The sentence is somehow ambiguous. Has it been hidden from scientists? Or from nature? Please clarify/rephrase.

How ubiquitous is this (endogenous?) mechanism?

"reducing basal leakage, increasing system output amplitude and dynamic range, and strikingly mitigating host growth inhibition induced by burden-causing regulatory proteins within the networks" sounds very cryptic. The authors should be more specific.

Reviewer #3:

Remarks to the Author:

"Synthetic protein-binding DNA sponge as a tool to tune gene expression and mitigate protein toxicity", Wan and Wang

The manuscript by Wan describes the characterization of operator and promoter decoys – so called DNA sponges - for their use as tools for the tuning and modulation of synthetic genetic circuits. While this approach has been used in the past, most notably in the updated repressilator by Potvin-Trottier et. al., their use has not been thoroughly investigated. Here the authors demonstrate the utility of DNA sponges by constructing plasmids containing repeated operator and promoter sequences and examine their effects on four different gene circuits involving biosensors, genetic amplifiers and genetic logic.

Overall, I think the paper is well written and logical. There is a detailed description of the circuits and behaviours contained in the supplementary information. The authors successfully demonstrate that DNA sponges could become a useful approach in the synthetic biology toolkit. Below I have some comments for improvement of the manuscript that need to be addressed.

Repetitive sequences and stability/recombination. It is mentioned on page 7 and the Discussion that there was cloning difficulty with plasmids with high numbers of repeats. This leads to the question of how stable these plasmids are. Was any work done to explore this, or can you show that the plasmids remain stable over suitable time scales?

On a related note, from the supplementary figures it looks like generally the noise in the circuits (as measured by the coefficient of variation) increases with the number of repeats in the DNA sponge. I know this is stated as interesting work for the future, but it is important consideration and could be due to recombination. It would be worth commenting more in the discussion, or even including a figure on noise in the main text (ie moving those supp figures into the main text).

Figure 1 would benefit from labelling the different sections (a, b, c etc) as it is difficult to follow which part of the figure are being referred to in the text.

Brophy and Voight mention decoy operators in their review on circuit design from 2014 so I think you should probably reference it: <https://www.nature.com/articles/nmeth.2926>

We thank all three reviewers for their constructive comments and suggestions to improve the manuscript (NCOMMS-20-23982A). Below we provide a point-by-point response (in blue) and list of changes (in red) to the reviewers' comments (in black).

Reviewer #1 (Remarks to the Author):

This is an interesting and important collection of work, demonstrating the utility of an elegantly simple method ("DNA sponges") in shaping the response curves of synthetic regulatory circuits. The ability to shape such curves effectively is vital to realizing synthetic biology's path towards being able to design genetic circuits with the kind of flexibility currently enjoyed in other branches of engineering. A substantial number of experimental tests of the system have been presented, making a significant case for the utility of the approach. Beyond the immediate application to circuits in *E. coli*, the general idea has more general relevance in other organisms, which I believe makes it of wide enough interest to appear here rather than in a more specialized journal.

We appreciate the reviewer for the overall positive comments on our work.

Some comments/questions:

- Would it be possible to clarify further what is happening in Figure 5? The best-growing (and thus least-burdened) condition is +20 LBS / +20 *P_{ecf11}*, but that condition also produces either the lowest or second-lowest level of GFP reporter output, if I'm interpreting panel (c) correctly. Level of reporter expression correlates with metabolic burden, so is the reduction in burden just a function of that effect? The fold activation is highest under those +20/+20 conditions, so perhaps the idea is to achieve the highest fold activation with the lowest required output level, or something similar? There's a reference in the Discussion to "sponging away sensitive transcription factors that tend to be burdensome at increased expression levels", but it wasn't entirely clear to me in the discussion of Figure 5 where/how this is happening. Some additional discussion, where Figure 5 is first introduced, of the mechanics underlying the improved growth rate (as a measure of decreased metabolic burden) would be helpful.

We thank the reviewer for the comment, and appreciate the reviewer's carefully review. The idea of employing dual sponge in this case is to achieve further improvement in output expression leakage, output induction fold as well as cellular burden. As noticed by the reviewer, the dual-layer sponge led to the highest induction fold as shown in Fig. 5e, but it also reduced the cellular burden as shown in Fig. 5f. Although the cellular burden seems to be related to the output expression level, the major cause of the burden observed was still the over expression of ECF11 protein. Here is our detailed explanation:

If we compare this ECF11-related genetic circuit with another genetic circuit that expressed the same amount of output GFP (e.g., Fig. 2g, LuxR-related circuit with Fluo./*A₆₀₀* at 20,000 a.u.), there is no obvious cellular burden observed for the latter circuit (Supplementary Fig. 3j), meaning the reporter expression was not the major cause of the cellular burden in the case of ECF11-related circuit. Instead, the low cell density of the host cells with ECF11-related circuit shows that ECF11 was the major cause of the burden. In addition, because the output expression is directly related to the available ECF11 within the cell, when the sponges are present (recruiting ECF11 molecules) there is an expected decrease in signal. Moreover, as the *P_{ecf11}* sponges only decoy the ECF11 without affecting its expression level while LBS sponges indirectly

decrease the ECF11 expression level, our results indicate that the metabolic burden is associated with both the expression level of ECF11 and its DNA binding. This is in line with a previous study (Rhodius et al. 2013) which suggested that the toxicity of ECF could derive from competition for native RNAP with host σ s and/or from aberrant gene expression.

Reference: Rhodius, V. A., Segall-Shapiro, T. H., Sharon, B. D., Ghodasara, A., Orlova, E., Tabakh, H., Burkhardt, D. H., Clancy, K., Peterson, T. C., Gross, C. a & Voigt, C. a. Design of orthogonal genetic switches based on a crosstalk map of σ s, anti- σ s, and promoters. *Mol. Syst. Biol.* **9**, 702 (2013).

Additional explanation has been added where Fig. 5 was first introduced as follows: “It is worth noting that the output expression was associated with the cell growth but was not the major cause of the reduced cell densities observed, since another genetic circuit with similar output level was not shown to be toxic to the host cells (**Fig. 2g, Supplementary Fig. 3j**), supporting that the cellular burden was largely caused by the expression of ECF11 itself. Due to the P_{ecf11} sponges decoy ECF11 without affecting its expression level while LBS sponges indirectly decrease ECF11 expression, our results suggest that the metabolic burden results from the high expression level of ECF11 and its binding activity. This is in line with a previous study which suggested that the toxicity of ECF could derive from RNA polymerase (RNAP) competition for native RNAP with host sigma factors and/or from aberrant gene expression³⁵”.

Reference:

35. Rhodius, V. A., Segall-Shapiro, T. H., Sharon, B. D., Ghodasara, A., Orlova, E., Tabakh, H., Burkhardt, D. H., Clancy, K., Peterson, T. C., Gross, C. a & Voigt, C. a. Design of orthogonal genetic switches based on a crosstalk map of σ s, anti- σ s, and promoters. *Mol. Syst. Biol.* **9**, 702 (2013).

- I found the structure of the Discussion section a bit unusual: Quite a bit of the section is spent introducing whole new types of results (measurements of the noise in gene expression, for example) that appear only in the Supplementary Information. It seems more appropriate to include those in the Results section.

We thank the reviewer for the suggestion. As also suggested by Reviewer #3, we now added a new result section “Synthetic DNA sponges are generally stable and have varied effects on expression noise” in the revised manuscript. The results of the noise measurements are moved to this section along with a new figure as follows:

“One challenge in synthetic circuit design is to minimise their effect on gene expression noise introduced and enhance the robustness of cellular response. In principle, decoying transcriptional factors may affect such noise significantly as it directly interferes with gene expression at transcriptional level, which has been shown to be the dominant source of gene expression noise³⁹. To investigate this, we evaluated how synthetic DNA sponges affected the noise of the output gene expression in our designed circuits (**Fig. 6c–d**). The noise was measured on the basis of robust coefficient of variation (C.V.) of the circuits’ output gene expression at single cell level. We found that noise was generally increased when DNA sponges were present for sponging off transcriptional repressors (i.e., TetR) (**Fig. 6d, Supplementary Figs. 5, 9, 11d**). For sponging off transcriptional activators (i.e., LuxR and ECF11), such effect turned out to be the opposite when low level activators were available for sequestration (**Fig. 6e, Supplementary Fig. 10g**), but remained for high activator levels in some cases (**Supplementary Figs. 6, 12i**). This suggests that the effect of protein decoying on gene expression noise may differ on a case-by-case basis and depend on the type and ratio of the decoyed proteins and decoys available.”

Reference:

39. Quarton, T., Kang, T., Papakis, V., Nguyen, K., Nowak, C., Li, Y. & Bleris, L. Uncoupling gene expression noise along the central dogma using genome engineered human cell lines. *Nucleic Acids Res.* gkaa668 (2020) doi:10.1093/nar/gkaa668.

Fig. 6: **c**, Schematics showing the study of noise effect of synthetic DNA sponge on synthetic circuit's output gene expression. The output fluorescence was compared at single cell level between the host cells with and without the presence of sponge. The noise was measured on the basis of robust coefficient of variation (C.V.) of the circuit's output gene expression. **d**, Robust C.V. of the output gene expression (left) and dose responses (right) of a two-layered AHL-responsive circuit (**Fig. 3a**) with or without tetO sponges from single cell assays. Values are mean \pm s.d. indicated by shading ($n = 3$). The cells were induced with 0, 0.02, 0.10, 0.39, 1.56, 6.25, 25, 100, 400 and 1,600 nM of 3OC₆HSL. **e**, Robust C.V. of the output gene expression (left) and dose responses (right) of a two-layered mercury-responsive circuit (**Fig. 3e**) with or without P_{ecf11} sponges from single cell assays. Values are mean \pm s.d. indicated by shading ($n = 3$). The cells were induced with 0, 0.008, 0.016, 0.031, 0.063, 0.125, 0.25, 0.5, 1 and 2 μ M of HgCl₂. Full profile of dose responses at single cell level and noise effect of other sponges are shown in **Supplementary Figs. 5, 6, 9, 10g, 11d, 12i**. Fluo., fluorescence. a.u., arbitrary units. Source data are provided as a *Source Data* file.

Reviewer #2 (Remarks to the Author):

In this manuscript, Wan & Wang show a novel approach to use DNA sponges to sequester prokaryotic transcription factors and shift regulatory equilibriums in cells.

Although the data is solid and well presented, the reviewer doubts that the data here has enough potential to be interesting to researchers in other related disciplines, and therefore justify publication in Nature Communications. Although, this work is surely of novelty and is relevant for scientists in the specific field, and should therefore be published in a more specialized journal.

We thank the reviewer for the comment, but we do not agree with the statement that our work is not suitable for Nature Communications. Nature Communications covers a wide range of research topics related to natural sciences including synthetic biology field, and we consider that our synthetic biology related work fits into these topics. We would also like to emphasize the comments from Reviewer #1 which agrees with our choice of publication in this journal: *"This is an interesting and important collection of work, demonstrating the utility of an elegantly simple method ("DNA sponges") in shaping the response curves of synthetic regulatory circuits"* and *"the general idea has more general relevance in other organisms, which I believe makes it of wide enough interest to appear here rather than in a more specialized journal."*

To improve the quality of the manuscript, some general comments:

It would be helpful for the reviewer to have line numbering so it would be easier for the reviewer to point to a certain position in the text.

We thank the reviewer for the suggestion. Continuous line numbers have now been added to the revised manuscript.

The reviewer was puzzled until the end of the introduction which type of organism this work is referring to. The authors should be clearer from the beginning that they are talking about prokaryotes, and their experiments are performed in *E. coli*. The reader should not be left thinking this work refers to eukaryotic systems until just before the results part begins.

We thank the reviewer for the suggestion. Although this work is focusing on prokaryotes, particularly *E. coli*, it could be applied to eukaryotes as well. We have clarified this at the beginning of abstract as follows: "Here, we investigated and repurposed a ubiquitous, **indirect endogenous** gene regulation mechanism ... to modulate target gene expression in *Escherichia coli*." and at the end of the introduction: **"Beyond the immediate application to circuits in *E. coli*, the synthetic DNA sponge-mediated regulation could also be applied to other prokaryotic and eukaryotic organisms."**

The authors never actually show binding of the targeted transcription factors to their DNA sponges. An experimental setup showing direct, physical interaction such as ChIP should be performed.

We thank the reviewer for the comment. The DNA sequences used for constructing the synthetic DNA sponges in this study are either the cognate promoters or operators of the selected transcription factor

proteins. Those motif sequences and their specific protein binding properties have been well studied by many other researchers (Potvin-Trottier et al. 2016, Rhodius et al. 2013, Uranowski et al. 2004), therefore we do not consider it necessary to show the physical binding here in this work. The ability of the transcription factor proteins to bind their cognate DNA motif sequences is demonstrated by the activity of the promoters themselves, in driving the expression of GFP reporter protein from the respective reporter plasmids. In addition, we have shown that the designed sponges could specifically modulate the regulated gene circuits' output expression, indirectly showing the binding of the proteins to the respective DNA sponges. Further, a previous study (Lee and Maheshri, 2012) has used ChIP assays to show the binding of their target tTA transcriptional factor protein to the synthetic tetO sponges, providing further supportive evidence that our selected transcription factor proteins should bind to their cognate promoters or operators-based sponges.

References:

Lee, Tek Hyung, and Narendra Maheshri. 2012. "A Regulatory Role for Repeated Decoy Transcription Factor Binding Sites in Target Gene Expression." *Molecular Systems Biology* 8 (1): 576. <https://doi.org/10.1038/msb.2012.7>.

Potvin-Trottier, Laurent, Nathan D. Lord, Glenn Vinnicombe, and Johan Paulsson. 2016. "Synchronous Long-Term Oscillations in a Synthetic Gene Circuit." *Nature* 538 (7626): 514–17. <https://doi.org/10.1038/nature19841>.

Rhodius, Virgil A, Thomas H Segall-Shapiro, Brian D Sharon, Amar Ghodasara, Ekaterina Orlova, Hannah Tabakh, David H Burkhardt, et al. 2013. "Design of Orthogonal Genetic Switches Based on a Crosstalk Map of σ_s , Anti- σ_s , and Promoters." *Molecular Systems Biology* 9 (702): 702. <https://doi.org/10.1038/msb.2013.58>.

Urbanowski, M. L., C. P. Lostroh, and E. P. Greenberg. 2004. "Reversible Acyl-Homoserine Lactone Binding to Purified *Vibrio Fischeri* LuxR Protein." *Journal of Bacteriology* 186 (3): 631–37. <https://doi.org/10.1128/JB.186.3.631-637.2004>.

Abstract:

What do the authors mean by "hidden gene regulation mechanism from nature". The sentence is somehow ambiguous. Has it been hidden from scientists? Or from nature? Please clarify/rephrase.

We apologize for the confusion. It is a hidden layer of gene regulation from nature, resulting from the competition trade-offs of resource sharing. In particular, it is an indirect protein binding-mediated regulation. We have changed "hidden" to "**indirect**" in the relevant sentence: "we investigated and repurposed a ubiquitous, **indirect endogenous** gene regulation mechanism from nature..."

How ubiquitous is this (endogenous?) mechanism?

We apologize for the confusion. The indirect protein binding-mediated regulation using protein-binding DNA decoy sites in the genome is commonly seen from endogenous gene regulatory networks. We have clarified it in the abstract: "we investigated and repurposed a ubiquitous, **indirect endogenous** gene regulation mechanism from nature..."

The protein decoying sites are highly abundant in many organisms' genomes and are important in cell development. See below relevant references:

Kemme, C. A., Nguyen, D., Chattopadhyay, A. & Iwahara, J. Regulation of transcription factors via natural decoys in genomic DNA. *Transcription* 7, 115–120 (2016).

Crocker, J., Preger-Ben Noon, E. & Stern, D. L. The soft touch: low-affinity transcription factor binding sites in development and evolution. in *Current Topics in Developmental Biology* vol. 117 455–469 (2016).

The mechanism behind this is essentially arisen from competition over limited molecular resources in cells. In particular, the DNA sponges decoy transcription factors from their cognate promoters. This has been included in the Introduction: “Both recent theoretical and experimental studies showed that DNA sponges regulate gene expression by decoying and therefore reducing the free-occupying transcription factors available for the target gene...”

“reducing basal leakage, increasing system output amplitude and dynamic range, and strikingly mitigating host growth inhibition induced by burden-causing regulatory proteins within the networks” sounds very cryptic. The authors should be more specific.

We thank the reviewer for the suggestion. We have now added more informative details to the sentence as follows: “reducing basal leakage **by >20-fold**, increasing system output amplitude **by >130-fold** and dynamic range **by >70-fold**, and strikingly mitigating host growth inhibition induced by burden-causing regulatory proteins **within the circuits by >20%.**”

Reviewer #3 (Remarks to the Author):

“Synthetic protein-binding DNA sponge as a tool to tune gene expression and mitigate protein toxicity”, Wan and Wang

The manuscript by Wan describes the characterization of operator and promoter decoys – so called DNA sponges - for their use as tools for the tuning and modulation of synthetic genetic circuits. While this approach has been used in the past, most notably in the updated repressilator by Potvin-Trottier et. al., their use has not been thoroughly investigated. Here the authors demonstrate the utility of DNA sponges by constructing plasmids containing repeated operator and promoter sequences and examine their effects on four different gene circuits involving biosensors, genetic amplifiers and genetic logic.

Overall, I think the paper is well written and logical. There is a detailed description of the circuits and behaviours contained in the supplementary information. The authors successfully demonstrate that DNA sponges could become a useful approach in the synthetic biology toolkit. Below I have some comments for improvement of the manuscript that need to be addressed.

Repetitive sequences and stability/recombination. It is mentioned on page 7 and the Discussion that there was cloning difficulty with plasmids with high numbers of repeats. This leads to the question of how stable these plasmids are. Was any work done to explore this, or can you show that the plasmids remain stable over suitable time scales?

We thank the reviewer for the comment. The *E. coli* TOP10 strain we were using for the sponge constructs cloning and testing is *recA* deficient (meaning this strain is deficient in DNA repair therefore has reduced occurrence of DNA recombination). However, there could be RecA-independent recombination occurring in the cells, particularly with a high number of DNA replicates on the plasmids. Therefore, we have tested the genetic stability of the sponge plasmids as suggested by the reviewer.

We selected the largest sponge from each sponge type, as well as two dual-layer sponges from **Fig. 4** and **5** (all tested sponges are listed in **Table 1**) to test the sponge stability. In brief, we cultured the strains with the sponge plasmids for 100 generations, extracted the plasmids every 20 generations, restriction digested the plasmids and compared the size of their DNA fragments with that from the 1st generation. We could not sequence the plasmids due to their large size and high numbers of DNA repeats. Although this approach would disregard any small changes in the sequences, we believe it is adequate to evaluate plasmid stability overtime.

The stability results are shown as gel images as follows (**Fig. 6b**), indicating the sponge plasmids we have successfully constructed are generally stable except the 40 × *P_{ecf11}* sponge which has been gradually lost in its host cells (**Supplementary Fig. 13**). This suggests that multi-layer sponges with lower number of repeats for each individual sponge might perform better in terms of genetic stability compared to a single-layer sponge with a high number of repeats.

Note that, it looks like the 40 × *P_{ecf11}* was totally lost in **Fig. 6b** but still remained some in **Supplementary Fig. 13**. It is because two different gel staining methods were used for the two experiments: post-staining for **Fig. 6b** and pre-staining for **Supplementary Fig. 13**. We found the DNA bands were weaker using the post-staining therefore did not show all the weak bands that were shown in pre-staining. The two different experimental approaches are explained in the corresponding Methods section.

Fig. 6: a, Schematics showing the workflow of the stability assay of synthetic DNA sponges. The cells harboring the sponge plasmids were cultured in 1 mL of medium in a 96-deep well plate and were diluted every 12 h for 5 days (corresponding to approximately 100 generations in total). Every 24 h, the cells were harvested for plasmid extraction, restriction digestion and the plasmids were analyzed by electrophoresis (see **Methods**). **b**, Image of a gel post electrophoresis showing the stability of synthetic single-layer DNA sponges of 320 × tetO, 40 × P_{tet2}, 80 × LBS, 40 × P_{ecf11} and dual-layer DNA sponges of 80 × LBS + 40 × tetO and 20 × LBS + 20 × P_{ecf11}. The sponge plasmids extracted from the host cells after 1 or 100 generations were compared. The plasmids from 20th, 40th, 60th and 80th generations and another two biological replicates are shown in **Supplementary Fig. 13**. The expected sizes of the selected DNA sponges are shown in **Table 1**. M, DNA marker. S, sponge. B, pSB3K3 backbone. The original image is provided as a *Source Data* file.

Supplementary Fig. 13: Stability assay of synthetic DNA sponges.

Image of gels post electrophoresis showing the genetic stability of synthetic single-layer DNA sponges of $320 \times \text{tetO}$, $40 \times P_{\text{tet}2}$, $80 \times \text{LBS}$, $40 \times P_{\text{ecf}11}$ and dual-layer DNA sponges of $80 \times \text{LBS} + 40 \times \text{tetO}$ and $20 \times \text{LBS} + 20 \times P_{\text{ecf}11}$ respectively. The sponge plasmids were extracted from the host cells after the 1st, 20th, 40th, 60th, 80th and 100th generations before restriction digestion. Arrows point to the positions of the digested DNA sponge constructs in each gel image. Three colonies of each sponge were tested as three biological replicates.

Relevant results description is included in a new result section “Synthetic DNA sponges are generally stable and have varied effects on expression noise”: “To support effective and long term use, synthetic DNA sponges should allow stable inheritance in their host cells. Given the large number of repetitive genetic elements present in the sponges, we proceeded to test the genetic stability of the sponges in their hosts. Although the *E. coli* TOP10 strain we used for sponge regulation study is *recA* deficient, there may be RecA-independent recombination occurring in the cells. To this end, we selected the largest sponge from each single-layer sponge type (i.e., $320 \times \text{tetO}$, $40 \times P_{\text{tet}2}$, $80 \times \text{LBS}$, $40 \times P_{\text{ecf}11}$) and two dual-layer sponges (i.e., $80 \times \text{LBS} + 40 \times \text{tetO}$ and $20 \times \text{LBS} + 20 \times P_{\text{ecf}11}$) to test their stability across 100 generations (5-day serial dilution) in their hosts (see **Methods**, **Fig. 6a**, **Supplementary Fig. 13**). We determined the plasmids stability

by analyzing DNA fragment sizes after restriction digestion, using gel electrophoresis. **Figure 6a** shows that most sponge plasmids are stable over 100 generations except the 40 × P_{ecf11} sponge which has been gradually lost in its host cells (**Supplementary Fig. 13**). This suggests that synthetic DNA sponges are generally stable and that multi-layer sponges with a lower number of repeats for each individual sponge may perform better in terms of genetic stability, when compared to a single-layer sponge with a high number of sponge repeats.”

Detailed experimental methods have been added as the new section “Sponge plasmids stability assays” to the Methods as follows:

“Sponge plasmids stability assays

E. coli TOP10 was transformed with selected sponge plasmids as listed in **Table 1**, plated on LB agar and grown for 17 h at 37 °C. A single colony of each sponge was inoculated into 1 mL terrific broth (TB, containing 12 g L⁻¹ tryptone, 24 g L⁻¹ yeast extract, 0.4% (v/v) glycerol, 17 mM KH₂PO₄, and 72 mM K₂HPO₄) in a 2.0 mL 96 Deepwell Plate (E2896-2110, StarLab). The liquid culture was grown for 17 h at 37 °C in the MB100-4A shaker incubator (Allsheng) with continuous shaking (1,000 rpm). The resulting cells were marked as their first generation. Then the cells were back diluted 1,000-fold into 1 mL fresh TB every 12 h for 5 days, which corresponded to approximately 10 × log₂ 1000 ≈ 100 generations with 10 generations for each transfer. The cells at 1st, 20th, 40th, 60th, 80th and 100th generations were harvested for sponge stability analysis. 50 µg mL⁻¹ kanamycin was used for all related experiments. Three colonies of each sponge were tested as three biological replicates.

Table 1: Details of selected sponges for plasmid stability assay.

DNA sponge	Length without plasmid backbone
320 × tetO	14.9 kb
40 × P _{tet2}	5.0 kb
80 × LBS	5.5 kb
40 × P _{ecf11}	3.7 kb
80 × LBS + 40 × tetO	7.4 kb
20 × LBS + 20 × P _{ecf11}	3.3 kb

To confirm the number of cell generations, the cell densities of the cell cultures before each transfer were determined by absorbance (A_{600}). 10 µL cell cultures were diluted into 90 µL fresh TB in a black 96-well microplate with clear bottom (655096, Greiner Bio-One), and the A_{600} was measured using the BMG FLUOstar plate reader. The background absorbance of the medium was determined from blank wells loaded with TB media and was subtracted from the readings of the other wells. The absorbance data were first processed using Omega MARS Data Analysis Software and then were exported to Microsoft Excel 2013 for data analysis. For each 12 h period, the number of cell generations were calculated using the following equation:

$$\text{Cell generations} = \log_2(A_t/A_0) \quad [\text{S4}]$$

where A_0 is the A_{600} value of the cell culture following dilution and A_t is the A_{600} value of the diluted cell culture post 12 h incubation. The relevant data are provided as Source Data.

To analyze the sponge plasmids stability, we purified the sponge plasmids and compared the size of the plasmids from the 20th, 40th, 60th, 80th and 100th generations with their 1st generation. Qiaspin Miniprep Kit (Qiagen) was used to purify the plasmid DNA, which was then quantified by DS-11+ Spectrophotometer (DeNovix). Unless otherwise indicated, 600 – 800 ng plasmid DNA was digested using FastDigestTM restriction enzymes *Xba*I and *Pst*I (ThermoFisher), and the digested DNA fragments were analyzed by electrophoresis using 0.5% (w/v) TBE (20-6000-50, Severn Biotech) agarose gel (16500500, ThermoFisher). 4 uL Quick-Load[®] 1 kb Extend DNA Ladder (NEB) was used as DNA size marker. The gel was run at 120 V for 70 min in a Wide Mini-Sub Cell GT Cell (Bio-Rad), followed by post-staining in SYBRTM Safe DNA Gel Stain (S33102, ThermoFisher) diluted 10,000-fold in 200 mL TBE at room temperature for 30 min with 40 rpm shaking incubation (on a ProBlotTM Rocker 25). For the tests shown in **Supplementary Fig. 13**, 550 – 850 ng of DNA were digested with *Xba*I and *Pst*I for 90 min at 37 °C before gel electrophoresis. The gels were pre-stained with SYBRTM Safe, and were ran at 400 mA constant current for 40 min. The gels were imaged using a Bio-Rad Gel Doc XR+ system with filter 1 and SYBR Safe mode. The gel images were acquired and adjusted by Image Lab software at 600 dots-per-inch resolution.”

On a related note, from the supplementary figures it looks like generally the noise in the circuits (as measured by the coefficient of variation) increases with the number of repeats in the DNA sponge. I know this is stated as interesting work for the future, but it is important consideration and could be due to recombination. It would be worth commenting more in the discussion, or even including a figure on noise in the main text (ie moving those supp figures into the main text).

We thank the reviewer for the comment. According to our sponge stability test results (shown above), the sponge plasmids are generally stable apart from sponge $40 \times P_{ecf11}$, but it did not cause high noise in its regulated gene expression (now **Fig. 6e**). Hence the increase of noise should not be due to the recombination. We found that the noise is generally increased when the sponge is decoying transcriptional repressors while the effect is opposite for low level transcriptional activators, therefore we suggest the noise effect may differ a case-by-case basis and could depend on the type and ratio of the decoyed proteins and decoys available.

As suggested, we have moved the text and figures that relevant to noise to a new result section “**Synthetic DNA sponges are generally stable and have varied effects on expression noise**”, and included discussion on the noise and recombination as follows:

In result section: “One challenge in synthetic circuit design is to minimise their effect on gene expression noise introduced and enhance the robustness of cellular response. In principle, decoying transcriptional factors may affect such noise significantly as it directly interferes with gene expression at transcriptional level, which has been shown to be the dominant source of gene expression noise³⁹. To investigate this, we evaluated how synthetic DNA sponges affected the noise of the output gene expression in our designed circuits (**Fig. 6c–d**). The noise was measured on the basis of robust coefficient of variation (C.V.) of the circuits’ output gene expression at single cell level. We found that noise was generally increased when DNA sponges were present for sponging off transcriptional repressors (i.e., TetR) (**Fig. 6d, Supplementary Figs.**

5, 9, 11d). For sponging off transcriptional activators (i.e., LuxR and ECF11), such effect turned out to be the opposite when low level activators were available for sequestration (**Fig. 6e, Supplementary Fig. 10g**), but remained for high activator levels in some cases (**Supplementary Figs. 6, 12i**). This suggests that the effect of protein decoying on gene expression noise may differ on a case-by-case basis and depend on the type and ratio of the decoyed proteins and decoys available.”

Reference:

39. Quarton, T., Kang, T., Papakis, V., Nguyen, K., Nowak, C., Li, Y. & Bleris, L. Uncoupling gene expression noise along the central dogma using genome engineered human cell lines. *Nucleic Acids Res.* gkaa668 (2020) doi:10.1093/nar/gkaa668.

Fig. 6: c, Schematics showing the study of noise effect of synthetic DNA sponge on synthetic circuit’s output gene expression. The output fluorescence was compared at single cell level between the host cells with and without the presence of sponge. The noise was measured on the basis of robust coefficient of variation (C.V.) of the circuit’s output gene expression. **d**, Robust C.V. of the output gene expression (left) and dose responses (right) of a two-layered AHL-responsive circuit (**Fig. 3a**) with or without tetO sponges from single cell assays. Values are mean \pm s.d. indicated by shading ($n = 3$). The cells were induced with 0, 0.02, 0.10, 0.39, 1.56, 6.25, 25, 100, 400 and 1,600 nM of 3OC₆HSL. **e**, Robust C.V. of the output gene expression (left) and dose responses (right) of a two-layered mercury-responsive circuit (**Fig. 3e**) with or without P_{ecf11} sponges from single cell assays. Values are mean \pm s.d. indicated by shading ($n = 3$). The cells were induced with 0, 0.008, 0.016, 0.031, 0.063, 0.125, 0.25, 0.5, 1 and 2 μ M of HgCl₂. Full profile of dose responses at single cell level and noise effect of other sponges are shown in **Supplementary Figs. 5, 6, 9, 10g, 11d, 12i**. Fluo., fluorescence. a.u., arbitrary units. Source data are provided as a *Source Data* file.

In discussion section: “Consistent with some previously reported theoretical studies⁴⁴⁻⁴⁶, our study showed that the effect of protein decoying on gene expression noise depended on the type and expression level of the decoyed transcription factors as well as the ratio of the decoyed proteins and decoys available. **Notably**, the increase in noise did not result from DNA recombination because our synthetic DNA sponges were shown to be generally stable in their host cells.”

Figure 1 would benefit from labelling the different sections (a, b, c etc) as it is difficult to follow which part of the figure are being referred to in the text.

We thank the reviewer for the comment. We have amended the figure and relevant figure caption as suggested as below:

Fig. 1: Schematic showing DNA sponge as a ubiquitous tool to tune gene expression and mitigate protein toxicity for gene circuit engineering.

a, Design of DNA sponges (in blue, top) to tune the response of a target gene circuit (in orange, bottom). A synthetic gene circuit typically comprises an input sensing module, an optional signal processing module (e.g., a transcriptional amplifier or inverter) and an output module for initiating physiological responses. Here in the input sensing module, a constitutive promoter (P_C) drives the expression of a receptor gene that responds to target ligands and regulates its cognate promoter P_R . In a typical signal processing module, the P_R drives the expression of a transcriptional activator (TA, blue solid circle) or repressor (TR, orange solid circle), which then controls its cognate promoter $P_{TA/TR}$ to express an output protein (e.g., GFP). The DNA sponge is designed to harbor one or multiple protein binding sites (rectangles) or cognate promoters (arrows) of the receptor or TA/TR in the target circuit. **b,c**, Diagrams showing the effects of the sponge regulation on the circuit's output response (**b**) and growth burden on the host (**c**). With the presence of different DNA sponges (green line/bar) for the receptor, TA or TR, the circuit's basal output expression can be reduced, leading to increased induction fold. In addition, the sponge can absorb excess toxic transcriptional regulators, leading to improved host cell growth.

Brophy and Voigt mention decoy operators in their review on circuit design from 2014 so I think you should probably reference it: <https://www.nature.com/articles/nmeth.2926>

We thank the reviewer for the suggestion. We have now included "Brophy, J. A. N. & Voigt, C. A. Principles of genetic circuit design. *Nat. Methods* 11, 508–520 (2014)" as ref 10 in the reference list.

Reviewers' Comments:

Reviewer #1:

Remarks to the Author:

I am satisfied with the responses to my questions.